# **Coupled Chemistry-Climate Effects from 2050 Projected Aviation Emissions**

Andrew Gettelman<sup>1</sup>, Chih-Chieh Chen<sup>1</sup>, Mark Z. Jacobson<sup>2</sup>, Mary A. Cameron<sup>2</sup>, Donald J. Wuebbles<sup>3</sup>, Arezoo Khodayari<sup>3,4</sup>

<sup>1</sup>National Center for Atmospheric Research, Boulder, CO, USA
 <sup>2</sup>Stanford University, Palo Alto, CA, USA
 <sup>3</sup>University of Illinois, Urbana, IL, USA
 <sup>4</sup>now at California State University Los Angeles, CA, USA

Correspondence to: Andrew Gettelman (andrew@ucar.edu)

- Abstract. Analyses of the climate effects of 2050 aviation emissions have been conducted with two coupled Chemistry Climate Models (CCMs) including experiments with coupled ocean models. The baseline 2050 aviation emissions scenario projects emissions ~5 times those in 2006. Simulations suggest a corresponding growth in the climate impact of aviation by 2050. Positive radiative forcing from contrails reaches +80mWm<sup>-2</sup>. Enhanced upper tropospheric and lower stratospheric ozone (O<sub>3</sub>) due to aviation nitrogen oxide (NOx) emissions causes a radiative forcing of +60mWm<sup>-2</sup>. Changes in methane
- (CH<sub>4</sub>) lifetime induced by aviation are estimated to cause -25mWm<sup>-2</sup> of radiative forcing in 2050. Simulations indicate that moderate changes in water vapor emissions from changes in combustion efficiency will not have significant forcing. Non-linear effects due to particles (black carbon and sulfur) included in these calculations suggest an important role for black carbon (BC) in increasing contrail cirrus ice crystal number, leading to net warming. Sulfur emissions brighten clouds and provide a net cooling, but this is dependent on uncertain background sulfur levels. Thus alternative aviation fuels with
- reduced sulfur and BC may alter the future climate impact of aviation, but the sign is dependent on specific processes represented and the background state. Regional perturbations due to contrail and particulate emissions may result in statistically significant regional surface temperature changes in coupled model simulations in areas near or adjacent to flight corridors, but significant signals only emerge after 20-50 years of simulation. Many regions with high regional aviation forcing do not experience net surface temperature changes because of advective rather than radiative driving of temperatures.
- Surface temperature signals are not significant globally even in long coupled simulations. Short-lived non-uniform aviation forcing will thus affect climate differently than uniform forcing in the coupled climate system.

#### 1. Introduction

Aviation has several important impacts on climate (Wuebbles et al., 2007; and Lee et al, 2009, Brasseur et al., 2008 & 2015). Commercial aircraft currently emit ~2% (Wuebbles et al., 2007) of global fossil-fuel related emissions of carbon dioxide (CO<sub>2</sub>). Aircraft also emit water vapor (H<sub>2</sub>O), nitrogen oxides (NO<sub>x</sub>), various organic gases, sulfur dioxide (SO<sub>2</sub>), black

carbon (BC) soot, other organic matter, and particulate sulfate. These chemicals may alter upper level cloudiness, atmospheric composition, and climate in several different ways.

For low enough temperatures and high enough exhaust humidity, aircraft emissions create condensation trails, or contrails
(Schmidt, 1941; Appleman, 1953). If the ambient air is supersaturated with respect to ice, linear contrails may persist for minutes to several hours (Minnis et al., 1998). The optical thickness of existing cirrus may be enhanced by the additional nuclei resulting from contrails that have aged due to spreading, shearing, and sublimating (Minnis et al., 1998). Aircraft emission effects on ozone (O<sub>3</sub>) and methane (CH<sub>4</sub>) lifetimes may also have radiative feedbacks on climate (Brasseur et al 2015).

Recent work summarized by Brasseur et al (2015) has shown significant 'non-linear' effects from aviation aerosols due to (a) aerosol absorption, termed 'direct' radiative effects and (b) modification of contrail and natural cloud microphysical properties by aerosols, or 'indirect' radiative effects. These effects may be as large or larger than the radiative effects of contrails themselves (Righi et al., 2013, Gettelman and Chen, 2013, Zhou and Penner, 2014). Aviation soot aerosol may

change their own properties as they age (Heymsfield et al 2010). Soot inclusions within and between contrail and background cloud particles also create a cloud absorption effect by enhancing contrail burnoff (Jacobson et al., 2013).

Many studies have explored the total non-CO<sub>2</sub> (short lived) effects of aviation radiative forcing. Lee et al. (2009) performed a review and meta-analysis and estimated aviation radiative forcing (RF) of 59 to 110 mWm<sup>-2</sup> for Ozone, -52 to -28mWm<sup>-2</sup>

for CH<sub>4</sub>, 37 to 55mWm<sup>-2</sup> for linear contrails, -12 to -18 for SO<sub>4</sub> (direct) and 9 to 13mWm<sup>-2</sup> for BC (direct). Chen and Gettelman (2016) recently performed simulations with one of the models used here and found significant indirect effects of aviation aerosols, especially sulfate, and RF increases by a factor of 5-7 from 2006 to 2050.

This manuscript evaluates and assesses the radiative forcing, ozone perturbation and surface temperature changes from projected levels of aviation emissions for 2050 in two advanced General Circulation Models (GCMs) with common scenarios for 2050 aviation emissions. The study focuses on simulations with coupled ocean, land and ice models. The goal is to determine statistically significant signals for small perturbations in coupled simulations. Previous studies have not been conducted with fully coupled "climate-response" models with small aviation forcing.

Section 2 describes the models, methodology and scenarios. Section 3 describes the results, and section 4 discusses them with a focus on comparing models. Section 5 provides conclusions and suggestions for future work.

2. Methodology

The climate-related effects described here include the radiative effects of contrails, the chemical effects of NOx on atmospheric concentrations of ozone  $(O_3)$  and methane  $(CH_4)$ , and the direct and indirect (or non-linear) effects of atmospheric particles resulting from emissions of BC, organic compounds, and sulfur species.

We use two different models with interactive chemistry and climate, described below.2.1 GATOR-GCMOM

GATOR-GCMOM (Gas, Aerosol, Transport, Radiation, General-Circulation, Mesoscale, and Ocean Model) is a one-way nested global-through-urban atmosphere-ocean-land model [Jacobson et al., 2011, 2013]. Resolution is 4° (latitude) x 5°

- (longitude) horizontal resolution with 68 vertical sigma-pressure layers from 0-60 km (0.219 hPa), including 15 layers from 0-1 km and 500-m resolution from 1-21 km. The model is initialized January 1, 2050 with 1°x1° reanalysis meteorological fields and ocean surface temperatures from January 1, 2012 [*GFS*, 2013] and run forward with no data assimilation. A single temperature profile was used to uniformly scale the ocean temperature relative to the surface, with temperatures at depth a constant ratio to the surface temperature. Thus the sub-surface ocean is not in balance as the stratification does not reflect
- spatial variations.

GATOR-GCMOM explicitly treats both the microphysical and radiative effects of aerosols on clouds and precipitation (Jacobson, 2012). Clouds, radiation and chemistry are coupled together, with feedbacks between them. GATOR-GCMOM treats the sub-grid evolution of aircraft particle and gas exhaust, including sub-grid contrail formation from each individual

- flight (30 million annually) [Jacobson et al., 2011, 2013; Whitt et al., 2010]. Non-aircraft anthropogenic emissions of NOx, SOx, NH3, CO, NMVOCs, CH4, BC, OC for 2050 are derived from the Fifth Assessment Report (AR5) inventory for 2050 assuming the Representative Concentration Pathway (RCP) 4.5 trajectory [Clarke et al., 2007]. In addition, for the long-lived greenhouse gases, CO<sub>2</sub>, N<sub>2</sub>O, CFCs, and HFCs, the EDGAR 2010 inventory [European Commission, 2014] extrapolated to 2050 are used. Biomass-burning gas and particle emissions are as in *Jacobson* [2012]. Natural emissions from lightning,
- soils, oceans, and vegetation are calculated as a function of meteorology as in Jacobson and Streets [2009].

The following global simulations are run for GATOR-GCMOM:

- 1) 2050 climate, 2050 non-aircraft anthropogenic emissions, and 2050 chorded aircraft emissions.
- 30 2) Same as (1) but with 2006 chorded aircraft emissions (but 2050 climate and 2050 non-aircraft anthropogenic emissions).
  - 3) Same as (1) but with no aircraft emissions.
  - 4) Same as (1) but with no BC in aircraft emissions.
  - 5) Same as (1) but with no SOx, NOx, CO, HC aircraft emissions.

- 6) Same as (1) but no contrail optical properties (contrail microphysics still occurred)
- 7) Same as (1) but with zero sulfur and 50% black carbon emissions (low-sulfur fuel scenario).

The simulations above have been run six complete years we show the first 5 years for comparisons. Results are substantially the same if year 6 is included. 20-year results for 2006 chorded emissions (with 2006 anthropogenic emissions and 2006 meteorology) have been documented by Jacobson et al. [2013].

Due to the high internal variability of the coupled climate system, long simulations are needed to achieve equilibrium. The GATOR-GCMOM simulations are transient simulations and are not in equilibrium. Long timescale feedbacks will not have

10 fully adjusting to the small aviation perturbation. This will limit interpretation of small surface temperature signals, but should not affect atmospheric responses to aviation ozone and aerosols. We recognize there are limits to this method. We use longer CESM simulations (see below) to understand the impact of these assumptions.

#### 2.2 CESM

Community Earth System Model (CESM, Hurrell et al 2012) with version 5 of the Community Atmosphere Model (CAM5) (Neale et al., 2010) uses a full gas-phase chemical mechanisms including tropospheric and stratospheric chemistry with 133 species and 330 photochemical reactions (Lamarque et al., 2012). CESM includes a detailed treatment of cloud liquid and ice microphysics (Morrison and Gettelman, 2008), including particle size distributions, a detailed mixed phase with a

- representation of water uptake onto ice (the Bergeron-Findeisen process) and ice supersaturation (Gettelman et al., 2010). Heterogenous freezing on dust and homogenous freezing are represented. Cloud microphysics is coupled to a consistent radiative treatment of ice clouds, and a 7-mode Modal Aerosol Model (MAM7) (Liu et al., 2012) with particle effects on liquid and ice clouds (Liu et al., 2012).
- Aviation emissions for water and other chemical species are put into CAM following Chen et al (2012) and using the AEDT emissions inventories (as described below). The contrail parameterization of Chen et al., (2012) is able to produce a reasonable spatial and seasonal distribution of contrails compared to observations (Chen et al., 2012). CESM has been used to estimate the present day impact of aviation emissions by Chen and Gettelman (2013). Gettelman and Chen (2013) estimated radiative effects of aviation aerosols. Chen and Gettelman (2016) estimated aviation RF in specified dynamics 30 simulations without chemistry.

2.2.1 CESM Specified Dynamics Simulations

To determine small radiative forcing perturbations, CAM specified dynamics (CAM-SD) simulations are used to eliminate meteorological noise. CAM-SD simulations use fixed meteorology (imposed pressure, winds and atmospheric and sea surface temperatures). Water, clouds, aerosols and chemical species are freely advected. The use of the SD model does not allow the feedback of aviation impacts or chemistry on the background meteorology.

Two sets of SD simulations were performed. The first set of SD simulations for determining contrail and aerosol effects are 10 years long, and use meteorology specified from a CESM coupled run. For future years, 4 sets of driving meteorology for each year (representing evolving forcing) were derived by running CAM with SSTs from a coupled simulation with RCP4.5 or RCP8.5 forcing for that year. No aviation emissions are included in the meteorology. Ensembles are created with a unit

- temperature perturbation. Then 4 pairs of simulations with and without aviation emissions are performed for each specified year using that meteorological forcing. Each of the 4 simulation pairs uses different meteorology, to test the variability of the results with respect to meteorology. The different years tested between 2006 and 2050 are: 2006, 2016, 2026, 2036 and 2050. For each of the 4 ensembles and each year (2006,2016,2026,2036,2050), we perform SD simulation pairs using RCP4.5 with different scenarios as outlined below.

SD simulations examining NOx effects are 5 year simulations are performed for the year 2050. These are basically the same as the 2050 CESM SD simulations, but run with more comprehensive chemistry. RCP4.5 boundary conditions of non-aviation NOx and volatile organic compounds (VOCs) were obtained from the IPCC RCP4.5 scenario for year 2050 (Van Vuuren et al., 2011). The monthly surface concentrations of longer-lived species, e.g., CO<sub>2</sub>, CH<sub>4</sub>, chlorofluorocarbons

(CFCs), and nitrous oxide (N<sub>2</sub>O), were specified as boundary conditions based on the IPCC RCP4.5 scenario. Meteorological fields to drive the SD simulation are also taken from a coupled CESM simulation for 2050.

The University of Illinois Urbana Champaign (UIUC) Radiaitve Transfer Model (RTM) was used offline to calculate the forcing associated with aviation NOx-induced short-term O<sub>3</sub>, in CESM while contrail effects are calculated in-line. The

25 UIUC RTM calculates the flux of solar and terrestrial radiation across the tropopause (e.g., Jain et al., 2000; Naik et al., 2000; Youn et al., 2009; Patten et al., 2011).

## 2.2.2 CESM Coupled Simulations

Coupled Simulations were also performed with CESM to explore potential surface temperature responses to aviation forcing. Coupled simulations feature a free running atmosphere coupled to a full dynamic ocean and sea ice model with a land surface model. Feedbacks between chemistry, clouds and meteorology occur. Simulations were performed for 50 years,

starting from a spun up control run of CESM for 2050 with RCP4.5. We only use the last 20 years for analysis, and allow 30 years for the model to adjust. These runs are closer to equilibrium than the GATOR-GCMOM simulations.

In order to evaluate the effect of transients in the simulations we also perform shorter CESM coupled experiments. These transient CESM simulations use the same ocean initial condition as GATOR-GCMOM: GFS reanalysis for SST for Jan 1, 2012. A single temperature profile was used to uniformly scale the ocean temperature relative to the surface, following GATOR-GCMOM. Salinity was not changed from the standard initial conditions in the RCP runs (the GATOR-GCMOM salinity was unstable in CESM). The biggest difference is the uniform temperature profile scaling used in the ocean. This run is analyzed to show the effect of a non-equilibrium state on the results from GATOR-GCMOM.

## 2.3 AEDT Emissions/Scenario

Simulations are conducted using a common set of emissions scenarios. Emissions used in this study come from the Aviation Environmental Design Tool (AEDT) emission inventory (Barrett et al., 2010; Olsen et al., 2012; Brasseur et al., 2015). The

- AEDT dataset is an hourly inventory of global aircraft emission mass of ten emission species over a 1° x 1° latitudelongitude mesh with a vertical spacing of 150 m in the year of 2006. 2006 data are chorded (individual flight tracks), and 2050 data are scaled and gridded. For GATOR-GCMOM, individual flight track 2050 aircraft emissions are obtained by extrapolating 2006 emissions in the same flight paths to 2050 by scaling the current year 2006 flight data by the ratio of 2050 to 2006 emissions. Emission indices from Barrett et al., (2010) are used for different species to determine particulate
- emissions. Both models use greenhouse gas concentrations in 2050 from the Representative Concentration Pathway 4.5 scenario (RCP4.5).

Different scenarios are listed in Table 1. The Baseline scenario assumes no operational or technology improvements. Scenario 1 has reduced fuel burn because of assumed technology improvements. Scenario 2 assumes the same fuel burn of Scenario 1 with an 'Alternative Fuel' (Alt Fuel) that has no sulfur and 50% less BC (soot) emissions. Scenario 3 assumes

- Scenario 1 with an 'Alternative Fuel' (Alt Fuel) that has no sulfur and 50% less BC (soot) emissions. Scenario 3 assumes Scenario 2 (reduced fuel burn with no sulfur and 50% BC) with +5% higher aviation H<sub>2</sub>O emissions from the engine exhaust. These scenarios were implemented in CESM. GATOR-GCMOM used 2050 baseline emissions, and a modified scenario 2 that assumes fuel burn of the Baseline scenario and an Alt Fuel version that has no sulfur and 50% less BC. This scenario (run only by GATOR-GCMOM) is called 'Scenario 2B'.

The baseline emissions scenario has a fuel burn in 2050 that is 5 times that of 2006 emissions. Over E. Asia, the 2050 emissions are nearly 8 times those of 2006. 2050 emissions changes are lower over the U.S. (2 times 2006) and Europe (4 times 2006). The reduced fuel burn scenario (Scenarios 1 and 2) is only 3 times 2006 emissions, with 4 times 2006 Asia, and 2 times 2006 for the U.S. and Europe.

The 2050 AEDT gridded inventory (Barrett et al., 2010; Brasseur et al., 2015) for Scenario 1 assumes a combined aircraft technology and operational improvement consistent with maintaining a 2 %/yr improvement in aviation system efficiency and a NO<sub>x</sub>-related technology improvement consistent with published ICAO/CAEP scenarios to 2036 extended to 2050 based on NASA N+3 and N+4 targets of "better than 75%" (Barrett et al 2010). The Baseline scenario does not include the

operational or technology improvements that were included in Scenario 1.

#### 2.4 Simulations

- GATOR-GCMOM was run for five model years with and without aviation emissions for baseline and Scenario 2B in a fully coupled configuration. Sensitivity tests were conducted as described in Section 2.1. CESM was run in both a specified dynamics (SD) mode, without feedbacks between emissions, chemistry and meteorology, to determine radaitive forcing (Section 2.2.1) and in a fully coupled mode to compare with GATOR-GCMOM (Section 2.2.2). CESM coupled experiments were run for 50 model years for longer-averaged statistics. CESM was run with baseline emissions for 2050 and all three
- scenarios.

## 2.5 Statistical Significance

Statistical significance was assessed using a 2-sided Student's t-test (e.g. Chervain and Schneider, 1976), estimating the standard deviation of annual means from the simulations with one degree of freedom for each year. Statistically significant differences are defined as those above 90% confidence. We also performed a paired sample t-test with autocorrelation on the difference of the annual means and got essentially the same significance results (with a more formal treatment of autocorrelation). The short nature of the GATOR-GCMOM simulations (5 years) means that the standard deviations may be large (and the degrees of freedom small). CESM simulations can be evaluated in a similar 5-year window, or with a longer

20-year window from the first 20 or last 20 years (31-50) of 50-year simulations to explicitly address the issue of short and transient simulations.

## 3. Results

First we present general results for fuel burn from the scenarios and illustrate the potential for contrail formation. We then examine the model-derived radiative forcing from contrails and other aviation emissions, including the radiative effects of chemical perturbations and the "non-linear" effects resulting from aircraft emissions of soot and sulfur. Next we examine the model-derived chemical effects on ozone and methane from NOx emissions. Finally, we analyze the resulting temperature

response from the aviation perturbations in 2050 based on coupled simulations. Note that the effect of alternative fuel is Scenario 2B v. Baseline for GATOR-GCMOM and Scenario 2 v. Scenario 1 for CESM.

# 3.1 Radiative Forcing in CESM

Radiative forcing for contrails and from aviation-related aerosols (both direct and indirect effects) were assessed in CESM. Top of atmosphere net downward solar plus thermal-infrared irradiance changes (a surrogate for radiative forcing but not the same quantity) were assessed in both CESM and GATOR-GCMOM. For clarity we focus on CESM radiative forcing.

- Figure 1 (adapted from Chen and Gettelman 2016) shows the evolution of the radiative forcing over time in CESM for different scenarios using SD simulations. The effects of water vapor only emissions (leading to contrail formation) are shown in Figure 1A, the effects of H<sub>2</sub>O + BC + SO<sub>4</sub> emissions in Figure 1B and the effects of BC+SO<sub>4</sub> emissions only in Figure 1C, as the residual of panel B minus panel A. Note that since H<sub>2</sub>O emissions for Scenario 1 and Scenario 2 are the same, their RF in Figure 1A is the same. Figure 1A shows clear increases in contrail cirrus radiative forcing to 2050. While
- global emissions grow by a factor of 5, contrail cirrus RF may grow by a factor of 7 (12 to 90 mWm<sup>-2</sup>). Note however that a warmer planet, using RCP4.5 meteorology (solid lines in Figure 1), lowers the resulting aviation RF relative to what it would be using 2006 meteorology (dashed lines in Figure 1), due to a reduction in contrail formation conditions at warmer temperatures (Chen and Gettelman, 2016). This assumes aircraft fly at the same pressure altitude in 2050 as today.
- RF grows faster than the emissions because the regional pattern of emissions is changing. Emissions are rising faster in regions like Asia, and in these regions the RF may be more sensitive to perturbations. Sensitivity may be higher either because there are fewer contrails in 2006 or the chance of forming contrails at flight altitudes is higher (since aircraft are always in the troposphere at low latitudes).
- Figure 1C shows the impact of aerosols in CESM. In the scenarios with significant sulfur emissions (Baseline, red and Scenario 1, green) there is significant decrease in TOA radiative forcing from aerosols (Figure 1C) that is anticipated to cause a near-surface cooling. Each line represents an average of 4 simulations (spread shown as error bars with symbols at the ends in Figure 1A and 1B).
- Eliminating sulfur in Scenario 2 (blue) and Scenario 3 (purple) eliminates the negative radiative forcing (Figure 1C). The net forcing is then positive (Figure 1B) and similar to figure 1A. Scenario 3 has a small amount of additional water vapor from aircraft. There is a small increase in contrail RF in 2050, on the order of 0-5% for a 5% increase in water vapor in Scenario 3, also seen in Figure 1B. The effect is small since most of the water in the contrails comes from the ambient atmosphere, not the fuel. The difference between scenario 2 and scenario 3 is not statistically significant.

Additional radiative forcing components are listed in Table 2, reprinted from Brasseur et al (2015). In particular, we have added CAM5 values of radiative forcing for 2050 for short-term effects of ozone, methane and water vapor for Scenario 2 (alternative fuels). We have also added available RF values for contrails and aerosols (Figure 1) from CESM-CAM5. Other

5 values are as described in Brasseur et al (2015). New values are in bold, and show small effects of alternative fuels on minor species radiative forcing. The reduction in sulfur and black carbon for alternative fuels in Scenario 2 changes short-term ozone RF by +2.2mWm<sup>-2</sup>, long-term ozone RF by -0.7 mWm<sup>-2</sup>, methane by -2.2 mWm<sup>-2</sup>, and water vapor by -0.5mWm<sup>-2</sup>. These effects in CESM are minor.

## 10 3.2 2050 Aviation NOx-induced effects on Ozone

#### 3.2.1. Effects on ozone

Figure 2A and B illustrate the simulated changes in zonally-averaged O<sub>3</sub> due to aviation emissions for (A) GATORGCMOM and (B) CESM. Note that CESM uses years 30-50 of a long coupled simulation while GATOR-GCMOM uses the first 5 years of a 5-year simulation. Results for CESM at different averaging periods are in the supplement, and are more similar between models.

Ozone radiative forcing is especially sensitive to the predicted ozone distribution in the Upper Troposphere and Lower 20 Stratosphere (UTLS) (Lacis et al., 1990). GATOR-GCMOM (Figure 2A) projects a peak increase in ozone at cruise altitude 20 between 45°N and 90°N of ~60 ppb which is not statistically significant. Effects at higher altitudes likely depend on 21 perturbations to the temperature gradient altering transport. However, due to variability of the stratospheric circulation they 22 are not significant. Significance is illustrated by the red dots in Figure 2 and other figures. Effects in CESM based on the first 25 years of a coupled simulation (Figure 5a) are similar in sign (~20 to 30ppbv) but also not significant as for GATOR-25 GEMON

25 GCMOM.

## 3.2.2 Effects on ozone with CAM5 using fixed meteorology fields

CAM5 was also run with fixed meteorology fields for 2050 (based on meteorology from free running simulations). The NOx-induced changes in tropospheric ozone are complicated by a short-term increase in O<sub>3</sub> concentrations associated with a positive forcing, and a long-term reduction of O<sub>3</sub> concentrations tied to the aviation induced methane decrease. The long term-reduction is associated with negative forcing (Stevenson et al., 2004). The short-term O<sub>3</sub> forcing is one of the major contributors to the overall aviation forcing and dominates the net O<sub>3</sub> forcing (Lee et al., 2009). Since CH<sub>4</sub> mixing ratios at the boundary layer are fixed, the calculated changes in O<sub>3</sub> concentration are the short-term changes.

Radiative forcing effects from these aviation perturbations are illustrated in Table 2 for each of the scenarios. Enhanced upper tropospheric and lower stratospheric ozone (O<sub>3</sub>), termed "short-term O<sub>3</sub> forcing", due to nitrogen oxide (NOx) emissions contributes  $\sim$ 60mWm<sup>-2</sup> of radiative forcing. Changes to the methane lifetime induced by aviation are estimated to induce a radiative forcing of -25mWm<sup>-2</sup> in 2050.

5 induce a radiative forcing of -25 mWm<sup>-2</sup> in 20

## 3.2.3 Alternative fuel effects on ozone

Figure 3A and B show the effect of using an alternative fuel on ozone. Both CESM and GATOR-GCMOM indicate very small effects on ozone by switching from traditional fuel to alternative fuel with reduced BC and no sulfur. Values are not significant. The aviation NOx-induced effect on ozone in the presence of sulfur and BC appears to be the same as for standard fuel.

#### **3.4 Temperature Effects**

GATOR-GCMOM and CESM were run coupled to an active ocean to estimate the temperature and surface temperature response to aviation emissions in 2050. The major motivation for running fully coupled climate simulations with GATOR-GCMOM and CESM is to assess the effects of aviation radiative forcing due to chemistry, contrails and aerosols on surface climate.

#### 3.4.1 Surface Temperature changes

Surface temperature changes are shown in Figure 4 for GATOR-GCMOM for the first 5 years for the effects of baseline emissions (Figure 4A) and the alternative fuel Scenario 2B (Figure 4B): very few of these differences pass a significance test

- in any coherent way. Similar results are seen for CESM simulations in the first 5 years (not shown). CESM is still adjusting to the forcing. CESM results over the years 31-50 for baseline (Figure 4C) and the effects of an alternative fuel (Figure 4D) show more significant patterns. Jacobson et al (2012) found with similar 20-year GATOR-GCMOM experiments that 16% of the planet with 'significant' temperature changes (>95% confidence level), including warming in the Arctic after a 20 year coupled experiment with present day aviation emission. In Figure 4 (and in Jacobson et al 2010), regions of significant
- surface temperature change are not precisely co-located with the largest forcing. For baseline emissions, net cooling from aviation sulfate in CESM causes significant Northern Hemisphere mid-latitude cooling (Figure 4C), consistent with the effect of the net negative radiative forcing for the baseline scenario in Figure 1b. The removal of sulfur in the Alternative Fuel Scenario 2 causes a reversal of this pattern (significant warming), associated with an increase in net radiative forcing (Figure 1B: scenario 1 to scenario 2).