# Peer review of "Coupled Chemistry-Climate Effects from 2050 Projected Aviation Emissions"

_Atmospheric Chemistry and Physics, 2017_

## Short Comment (SC1) · 5 May 2017

Some comments on A. Gettelman et al. "Coupled chemistry climate effects from 2050 projected aviation emissions"

(This is not meant as a full scale review but rather a collocation of ideas that occurred to me when reading through the paper. Nevertheless, I am confident that the comments express well-founded criticism.)

The present paper fails to discuss evidence from previous work to an extent that makes it difficult to understand what is actually added here to current knowledge on the subject. Besides other papers I am particularly referring to Huszar et al. (2013), a basic study of future aviation impacts that the authors appear to have overlooked.

1) The results given in section 3.1 all in all look scientifically and statistically plausible, yet they have apparently been presented before (Brasseur et al., 2015; Chen and Gettelman, 2016). However, I notice an inconsistency with respect to the contrail cirrus RF estimate for the 2006 baseline scenario between Table 2 (17 mW/m$^2$) and Figure 1A (12 mW/m$^2$). This is rather relevant, as the puzzling evidence that contrail cirrus RF increase more strongly over the years than fuel consumption would vanish, if the Table 2 value were taken as the starting value.
Recently, Forster et al. (2016) came up with a study indicating that radiative flux differences derived from free-running fixed SST simulations (I guess that's what "RESTOM" indicates in Figure 1) should amount to at least 100 mW/m$^2$ in order to reach sufficient statistical significance levels. In case of nudged simulations (resembling the specified dynamics simulations in the present papers) the threshold value may reduce to 10% (Forster et al., 2016, p. 13), which appears to be consistent with the error bars in Figure 1. However, the error bars are clearly overlapping between the different scenarios at all time slices, indicating that the scenarios are statistically indistinguishable.

2) I find the ozone pattern difference presentations from Fig. 2 a,b, Fig. 3 a,b, Fig. 5, Fig. 6, rather pointless. While they suggest large areas of statistical significance (for CESM almost over the whole troposphere), this remains unconvincing as contour lines are largely missing (those that *are* shown are mainly referring non-significant structures). Figure 8 of Huszar et al. (2013) offers a more satisfactory description, clearly indicating that patchy stratospheric response patterns are insignificant, despite showing higher concentration difference values compared to the troposphere. It may, hence, be worthwhile to display relative differences, as in many earlier papers (e.g., Grewe et al., 1999) dealing with free running chemistry climate model simulation results.

3) The severe problems to assess the (statistical and physical) significance of temperature response patterns simulated from aviation effects have been reported before (e.g., Rap et al., 2010, Fig. 1a; Huszar et al., 2013, Fig. 10, Fig. 12). A point-by-point hypothesis test suggesting statistical significance in strongly confined regions may well turn out to be unfounded, if spatial correlation is accounted for. (Chaotic negative and positive side-by-side differences, as obvious in Fig. 2 c,d, Fig. 3 c,d, are always raising suspicions in this respect. I notice coherent regions of significance only in Figs. 3d, 4d, and 6f shown here.) Significant temperature response is more easily established for global means (Huszar et al., 2013, Fig. 9), but these are bypassed in the present paper. Sometimes, more sophisticated (multivariate) statistical tests have proved helpful to establish pattern significance (e.g., Sausen et al., 1998).

4) In section 4.1.2 much text is devoted to allegedly large effects of aviation black carbon emissions without showing any results. To me this is absolutely unconvincing as to underpin what is suggested by Figure 1b.

5) Given the general lack of statistical significant simulation results, I find large parts of the concluding section to be insufficiently covered by the results. In my opinion, the simulation strategy followed in this paper is only of very limited value for establishing reliable evidence on the relative importance of individual components in forcing a net aviation climate impact. Even in Huszar et al. (2013) statistical noise has made interpretation of their results problematic and I fail to notice any progress on this in the present paper.

Adding on my main comments, I find the present paper to be written in a rather confusing manner. For example, the description of the simulations is scattered over three different sections (2.1, 2.2, 2.4) and it is not sufficiently recalled in the results section, which of the simulations are actually discussed. A special subsection (3.4.3) is dedicated to alternative fuel effects, yet those are partly addressed already in subsection 3.4.1.

References:
Forster, P.M., et al., 2016: Recommendations for diagnosing effective radiative forcing from climate models for CMIP6, J. Geophys. Res. 121, 12460-12475.
Grewe, V., et al., 1999: Impact of future subsonic aircraft NOx emissions on the atmospheric composition, Geophys. Res. Lett. 26, 47-50.
Huszar, et al., 2013: Modeling the present and future impact of aviation on climate: an AOGCM approach with online coupled chemistry, Atmos. Chem. Phys. 13, 10027-10048.
Rap, A., et al., 2010: Estimating the climate impact of linear contrails using the UK Met Office climate model, Geophys. Res. Lett. 37, L20703.
Sausen, R., et al., 1998: Climate impact of aircraft induced ozone change, Geophys. Res. Lett. 24, 1203-1206.

---

## Referee Comment (RC1) · W.J. Collins (Referee) · 16 May 2017

This paper calculates the non-CO2 climate impacts of future aircraft emissions using two very different models. There are some reasonably robust changes in radiative forcing from the CESM model, but the changes in temperature are not found to be robust. Overall it is not clear what the new findings coming from this study are. The work is publishable, but more thought needs to be given to the overall messages if this paper is to be of interest to the community.

The two models are set up differently which makes it very difficult to draw any useful conclusions from their comparison. In particular the absence of radiative forcing data from GATOR-GCMOM means that the conclusions in section 5 are mostly speculative. Ideally both models would also have run fixed-SST experiments to categorize the rapid

responses.

In general the text doesn't flow very well, with many short paragraphs that don't seem to connect. It would help to understand the messages better if there was a logical chain of argument that could be followed.

Specific points:

Page 2, line 11: I'm not sure describing the aerosol effects as 'non-linear' is a helpful term. None of the aircraft impacts are strictly linear.

Page 2, line 23: "and RF increases by . . .". This clause doesn't seem to sit with the rest of the sentence.

Page 5, line 9: What is meant by "Ensembles are created with a unit temperature perturbation"?

Page 5, line 25: I presume the fluxes are taken at the tropopause because there is no stratospheric adjustment? Does this give equivalent results to a RTM with stratospheric adjustment?

Page 7, line 32: "non-linear" isn't the right term.

Page 8, lines 20-24: Surely the model can tell you whether there are fewer present day contrails, or a higher change for forming contrails?

Page 8, lines 30-34: Presumably the aerosol emission affect the contrails as well, which should be discussed here. Why are the differences between scenarios 2 and 3 not statistically significant? If they are run with specified dynamics the meteorology should be the same and hence no (or very little) variability. I don't understand why the effect of water vapour emissions is so small. According to figure 1A the water vapour alone has a huge forcing.

Page 9, line 7. The effect of alternative fuels here doesn't seem the same as the difference between the lines 2050-S1 and 2050-S2 in table 2. In particular in the table

the effect on O3-S is 12.0 mW/m2 which seems large. The authors should explain how the changes in sulfur and BC cause such a large change in ozone.

Page 10, lines 1-5. Which scenarios do these forcings come from?

Section 3.2.3: Given that it isn't expected that the aerosols affect ozone I suggest this (very short) section isn't needed, nor are figures 3A and B, or 6 A,B,C.

Section 3.4: Should this be numbered 3.3?

Section 3.4.2: The arguments in this section needs to be made clearer. Figure 2D needs to be the 5-year result from CESM for like-for-like comparison with GATOR. The time evolution needs to be discussed in relation to rapid responses to composition followed by slower responses to SST evolution. It is not clear what the message of the second two paragraphs is. Description of the physics should be moved to section 2, unless the authors are specifically contrasting the different effects of the physics in GATOR and CESM.

Page 12, line 10. "...contrail radiative forcing dominates..." I don't see why this is true, don't scenarios 1 and 2 have similar contrails?

Page 13, line 1. Where does GATOR show a small warming? In figure 4 it cools.

Page 13, lines 11-13: I didn't see the relevance of this disconnected paragraph on Righi et al.?

Page 13, line 15-16: This disconnected paragraph needs to be moved somewhere else as part of a logical train of argument.

Page 13, lines 20-34: Much of this is model description which could be moved to section 2. Earlier (page 12, line 30) the BC and sulfate are described as externally mixed in the exhaust, but here they are described as internally mixed.

Page 14: These short paragraphs disrupt any flow of argument. What is the message of this section?

Page 15, line 11-12: The forcing in GATOR needs to be shown to back this up.

Page 15, lines 17-20: The difference between the baseline (-0.11K) and the AltFuel (+0.1K) is 0.2K. While this may not be statistically significant due to the length of the simulations, this is not a negligible difference compared to the 1.5-2.0K Paris recommendations.

Section 5: The paragraphs in this section tend to be short and unconnected which makes it difficult to pull out the important messages of this study.

Figure 2: A different set of contour levels is needed to show the ozone changes.

Figures 3, 5, 6: The ozone panels don't add information here.

---

## Referee Comment (RC2) · Anonymous Referee #2 · 17 May 2017

The paper presents results from two climate-chemistry-ocean models on the climate impact of future aviation. In particular the paper discusses the climate impact of aviation emissions on climate temperature changes.

Certainly, the topics are interesting and important and deserve careful investigations. An ocean-atmosphere coupled model with a high quality model simulating the fate of the aircraft emissions is essential for computing aviation climate change.

Unfortunately, the paper is insufficient in many respects. The material presented, though not irrelevant, does not add enough new insight and results to the existing literature. This topic deserves a far deeper investigation and a technically better paper.

The paper reports quantitative values of the radiative forcing (RF) values for various aviation emissions and effects. The majority of these results are taken from earlier

publications and given here with little or no discussion on the ranges of validity and discussions on the differences to results from other studies. Additions concern specific scenarios.

Then the study reports various "nonlinear effects" from BC, including strong solar radiation absorption by BC from engine exhaust in cirrus, sulfur brightening of low level clouds, regional disturbances, advection effects, and surface temperature response.

The simulations were performed with two different models. Here too, a large part of the results was published earlier (in the various cited papers by Jacobson, Chen and Gettelman). In fact, the paper has strong overlap with Brasseur et al. (2016), Jacobson et al. (2013), and Chen and Gettelman (2016), even repeating some of the numerical values in the tables and one of the already published figures. The two models disagree in many respects, and the discussion mentions possible reasons, but the discussion remains qualitative and does not present new convincing evidence explaining the differences clearly.

Not surprisingly, the authors did not find statistically reliable results in this respect. This was to be expected, as discussed in other studies, because the disturbances are small compared to the inherent climate noise. For this reason, other authors either use enhanced disturbances or follow the idea of climate response models, which are quasi linear in the disturbances, with model parameters (inertia or heat capacities and climate sensitivities) fitted to full climate model studies with enhanced disturbances. The possible inaccuracies of such approaches because of inherent nonlinearities are unavoidable. It would have been interesting to see how linear or nonlinear the model responses are (see Rind et al., 2000).

As the authors mention themselves, none of the climate change results on regional temperature and ozone changes and surface temperature changes is strictly statistically significant. Even the global mean surface temperature changes remain in the statistical noise. (This seems to revise some earlier conclusions form the same data;

e.g., Jacobson et a. 2013).

The patterns of the simulation results do not look convincing. There is little systematic pattern in the mean responses. Many of the results just look random. Fig. 2b and 4 are examples.

I am not sure, whether the significance tests are reliable because based on local tests; see Wilks, D. S. (2016), "The stippling shows statistically significant grid points": How research results are routinely overstated and overinterpreted, and what to do about it, Bull. Amer. Meteorol. Soc., 97, 2263-2273, doi: 10.1175/BAMS-D-15-00267.1. This should be discussed.

The suitability of the models for this study is not sufficiently justified. Since a global model with very coarse horizontal resolution ($4 \times 5$ degrees) cannot resolve plume dispersion of NOx and neither line-shaped contrail formation nor their merging into contrail cirrus, the results rely highly on the validity of the subgrid scale (SGS) models used. The present method gives no details on these SGS models but refers to previous publications. When looking to some of the previous publications one finds a lot of ad-hoc simplifications. I agree that simplifications are unavoidable, but I see a lack of principle justification (e.g., on plume cross-sections), reflection of recent insight and data, and lack of validation of the SGS models with the observations that are now available from various studies. The paper does not discuss the degree of agreement or disagreement with related contrail studies, e.g., in respect to contrail ice water content, optical depth, life times, cross-sections etc.. Hence it is unclear of how good the SGS models are and how much the results change when the SGS model is changed.

The assumption that the radiative properties of contrail cirrus are the same as those of normal cirrus clouds is highly questionable and has been overcome in other studies. We know since Minnis et al. (1998) that contrails remain observable at ages larger 10 h. Many further measurement results have been presented on this since then. Many findings indicate that aged contrails differ significantly from other cirrus. In particular
they often contain higher concentrations of small ice particles, with impact on optical properties, sedimentation and life times.

A prerequisite for contrail cirrus simulations is the suitability of the simulations of ice supersaturation (and temperature). See Irvine, E. A., and K. P. Shine (2015), Ice supersaturation and the potential for contrail formation in a changing climate, Earth Syst. Dynam., 7, 555–568, doi: 10.5194/esd-6-555-2015. It would be important to show how good the present models resolve temperature and ice supersaturation along aircraft flight tracks, e.g., compared to qualified reanalyzes or to in-situ measurements.

With respect to NOX and O3, I miss a discussion of the dispersion of NOx etc. from aircraft engines to grid scale which is known to cause nonlinear O3 changes since early studies in the 1990's, depending on the dilution model assumed. See, e.g., Paoli, R., D. Cariolle, and R. Sausen (2011), Review of effective emissions modeling and computation, Geosci. Model Dev., 4, 643-667, doi: 10.5194/gmd-4-643-2011.

The question whether aviation NOx emissions cause a positive or negative or zero RF is still under debate. See Pitari et al. (2016), Radiative forcing from aircraft emissions of NOx: model calculations with CH4 surface flux boundary condition, Meteorol. Z., 23, doi: 10.1127/metz/2016/0776. The CH4 surface boundary condition seems to matter. The present study uses prescribed CH4 at the surface which likely has consequences for the results.

Work from other teams is hardly mentioned. Differences in the results between this study and other studies (e.g. for NOx induced O3 and CH4 changes or contrail RF) are neither mentioned nor discussed.

As mentioned in a comment by M. Ponater, the studies by Olivié et al. (2012) and Huszar et al. (2013) are related to this work, and should have been discussed.

The results and conclusions are not always clearly presented. One example: The authors relate (in the abstract and the conclusions) the non-local surface temperature

signal from local radiative forcing to advection. The assumption that advection is the reason for nonlocal behavior is reasonable and not fully new (Ponater et al, Ann Geophys., 1996; Shindell and Faluvegi: Climate response to regional radiative forcing during the twentieth century, Nature Geosci., 2, 294-300, doi: 10.1038/NGEO473, 2009; see also Rind et al., 2000). However, this paper does not bring any new argument to support this conclusion in this paper, except that the (noisy) temperature patterns exhibits downstream shifts. Is that worth mentioning as a new finding in the abstract? I think, this needs more analysis.

The authors claim that heat absorption by BC from aviation is large enough to cause strong warming in contrails. They refer to Liou et al. (2013) in this respect. There is no doubt that BC does change absorption when present in sufficient amount. The quantitative results depend on the assumed BC mass (and effective sizes) of the soot and the mass of ice particles (and their sizes) and the fraction of ice particles containing soot particles. It would be important to check the mass budget of the BC in ice particles and see if this mass budget is consistent with the aviation BC emissions, the lifetime of cirrus and aerosol sinks. For example one could analyze from the model results the total ice mass in the cirrus clouds regionally or globally, convert that to cross-sections, and could compare this with the total BC mass from aviation in the same clouds. I would not be surprised if the mass or area fraction of BC turns out to be by far smaller than that for cirrus ice.

In spite of parallel studies by Righi et al. and Gettelman et al., I am not convinced that aviation sulfur emissions change low level clouds in any significant manner. I miss a careful and critical discussion of the amount and concentrations of cloud condensation (CCN) particles of reasonable sizes which could be contributed by aviation compared to the many other sources for CCN. Again, one could compute from the model results related statistics. To my understanding, most of the CCN in stratus clouds come from non-sulfur sources. I do not know of any single measurement showing that aircraft could indeed change water clouds by sulfur emissions. So, to me, this effect appears

to be purely hypothetical.

---

## Author Comment (AC1) · 20 Jul 2017

Replies to Reviewers

We sincerely thank the Reviewers for their effort. We appreciate this is a difficult manuscript. We are trying to figure out the appropriate way to present additional work (related to previous work). That requires a bit of repetition, which we have tried to reduce. And we are trying to appropriately present the statistics of our simulations with small perturbations: it is difficult to find a statistically significant signal in surface temperature from aircraft emissions in coupled simulations: it is smaller than the climate noise in most cases. We have however re-run our statistics using the false discovery rate (FDR) approach suggested. This provides we think a more robust result. And

we have now found regionally significant temperature results, and global temperature changes that are barely significant. We thank the reviewers for this suggestion and the pointer to the new method. We think this significantly strengthens the conclusions, and we thank the reviewers for pointing us to these additional tests.

In addition, we have modified the text to try to better discuss model processes and our model comparisons. It is not easy to compare the models, which are very different in construction. We are doing our best because this was not a controlled experiment. The coupled nature of the models is why this is has not been done before. Nonetheless, we have performed tests to bring the models together by testing GATOR-GCMOM assumptions in CESM, and we think this does help understand the processes and assumptions responsible for model differences. This is also a unique aspect to the study.

As noted above, we have redone the statistical tests to be more robust at the suggestion of the reviewers. We have also modified the presentation of the Ozone figures, and added further process discussions of the mechanisms leading to the radiative forcing changes we describe with additional analysis.

We have also rewritten the conclusions for clarity to make them flow better, and better state the revised statistical results.

We think this revised manuscript will answer the reviewers' concerns, and we hope the spirit of this attempt now comes through with better focus in the revised manuscript.

Reply to Dr. Collins

This paper calculates the non-CO2 climate impacts of future aircraft emissions using two very different models. There are some reasonably robust changes in radiative forcing from the CESM model, but the changes in temperature are not found to be robust.

» With slightly more attention to the statistics and a better formulation as described

above, we think now we can state some regionally significant results, and we now characterize the global changes as 'barely significant' which we think is true (it passes our 95% significance test, but barely).

Overall it is not clear what the new findings coming from this study are. The work is publishable, but more thought needs to be given to the overall messages if this paper is to be of interest to the community.

» Clarified. The main point is to summarize the result of fully coupled experiments. This is now stated explicitly in the introduction, abstract and conclusions. We think we have a more robust and coherent result now, and have rewritten the conclusions significantly

The two models are set up differently which makes it very difficult to draw any useful conclusions from their comparison. In particular the absence of radiative forcing data from GATOR-GCMOM means that the conclusions in section 5 are mostly speculative. Ideally both models would also have run fixed-SST experiments to categorize the rapid responses.

» Both models have run fixed SST and even fixed meteorology experiments to explore the rapid responses. This is now noted in the text more explicitly. We would not say the GATOR-GCMOM results are speculative, rather that they are not significant given the large variability in 5 years of coupled simulation.

In general the text doesn't flow very well, with many short paragraphs that don't seem to connect. It would help to understand the messages better if there was a logical chain of argument that could be followed.

» We have now tried to summarize the sections more and provide more discussion of this logical chain. We have also reorganized the text a bit, and tried to remove some of the shorter paragraphs

Specific points: Page 2, line 11: I'm not sure describing the aerosol effects as 'non-linear' is a helpful term. None of the aircraft impacts are strictly linear.

[Figure]

» Deleted 'non-linear'

Page 2, line 23: "and RF increases by : : :". This clause doesn't seem to sit with the rest of the sentence.

» Clarified (specific numbers mentioned)

Page 5, line 9: What is meant by "Ensembles are created with a unit temperature perturbation"?

»Clarified. This is the method used to initialize different ensemble members, with a round off level temperature perturbation.

Page 5, line 25: I presume the fluxes are taken at the tropopause because there is no stratospheric adjustment? Does this give equivalent results to a RTM with stratospheric adjustment?

» Yes, there is no stratospheric adjustment with specified dynamics. It is not wise to try to compare to RF estimates with stratospheric adjustment when dealing with forcing in the UTLS around the tropopause, as with aviation.

Page 7, line 32: "non-linear" isn't the right term.

» Removed non-linear (now removed from the whole manuscript).

Page 8, lines 20-24: Surely the model can tell you whether there are fewer present day contrails, or a higher change for forming contrails?

» There are fewer present day contrails, so there may be higher sensitivity. We have clarified this paragraph.

Page 8, lines 30-34: Presumably the aerosol emission affect the contrails as well, which should be discussed here. » Noted.

Why are the differences between scenarios 2 and 3 not statistically significant? If they are run with specified dynamics the meteorology should be the same and hence no (or

very little) variability. I don't understand why the effect of water vapour emissions is so small. According to figure 1A the water vapour alone has a huge forcing.

» Clarified. The 'water vapor' in figure 1 is due to emissions of aviation water vapor causing contrail formation. The difference from Scenario 2 to Scenario 3 is a 5% increase in aviation water vapor emissions, which does not affect things substantially. The contrails mostly contain ambient humidity, not humidity from the engines.

Page 9, line 7. The effect of alternative fuels here doesn't seem the same as the difference between the lines 2050-S1 and 2050-S2 in table 2. In particular in the table the effect on O3-S is 12.0 mW/m2 which seems large. The authors should explain how the changes in sulfur and BC cause such a large change in ozone.

» The reviewer is correct. The paper mis-stated the table value for Short term Ozone effect. This is larger for short term ozone effect. The impact of alternative fuels is ∼15%, which is not that large in relative terms. This is probably an 'indirect effect' on ozone that arises from changes in UTLS temperature induced by BC reductions, but the result may not be significant given the small changes in ozone in Figures 3A and B. Noted in the text.

Page 10, lines 1-5. Which scenarios do these forcings come from?

» Clarified (scenarios in Table 1).

Section 3.2.3: Given that it isn't expected that the aerosols affect ozone I suggest this (very short) section isn't needed, nor are figures 3A and B, or 6 A,B,C.

» For completeness, we left the section in. And the figures are necessary for the comments about Ozone RF above.

Section 3.4: Should this be numbered 3.3?

» Yes. Corrected throughout.

Section 3.4.2: The arguments in this section needs to be made clearer.

Figure 2D needs to be the 5-year result from CESM for like-for-like comparison with GATOR.

» It is shown in Figure 5, but we choose to focus the discussion on the 31-50 year period because that is where statistical significance is. Figure 5 is referenced here.

The time evolution needs to be discussed in relation to rapid responses to composition followed by slower responses to SST evolution.

» Noted and clarified.

It is not clear what the message of the second two paragraphs is.

» The second paragraph has been shortened and clarified. It's purpose we think is well described by the topic sentence "the reduction of BC in the Alt fuel scenario (Figure 3C) reduces warming relative to the baseline". The 3rd paragraph has been focused and shortened and it's goal is to note that BC is the most important component in GATOR-GCMOM.

Description of the physics should be moved to section 2, unless the authors are specifically contrasting the different effects of the physics in GATOR and CESM.

» The discussion here of the physics has been shortened.

Page 12, line 10. ": : :contrail radiative forcing dominates: : :" I don't see why this is true, don't scenarios 1 and 2 have similar contrails?

» Clarified. They do have similar contrails. The indirect sulfate aerosol cooling has been removed.

Page 13, line 1. Where does GATOR show a small warming? In figure 4 it cools.

» Clarified.

Page 13, lines 11-13: I didn't see the relevance of this disconnected paragraph on Righi et al.?

» Merged with above paragraph.

Page 13, line 15-16: This disconnected paragraph needs to be moved somewhere else as part of a logical train of argument.

» Removed (discussed in conclusions).

Page 13, lines 20-34: Much of this is model description which could be moved to section 2.

» Actually we think it belongs here. Section 2 references the parameterizations, but here we directly discuss the aspects of the parameterizations that matter for the differences, which is appropriate for the discussion section.

Earlier (page 12, line 30) the BC and sulfate are described as externally mixed in the exhaust, but here they are described as internally mixed.

» Clarified (CESM does treat internal mixtures).

Page 14: These short paragraphs disrupt any flow of argument. What is the message of this section?

» The section is designed to analyze differences in the BC results between the models, and to describe analysis that bring CESM results more in line with GATOR-GCMOM. We have merged several paragraphs and moved the last paragraph to the conclusions. This makes sure the section wraps up with a strong conclusion comparing to previous work.

Page 15, line 11-12: The forcing in GATOR needs to be shown to back this up.

» Radiative forcing was not calculated from GATOR-GCMOM. The sentence has been rephrased.

Page 15, lines 17-20: The difference between the baseline (-0.11K) and the AltFuel (+0.1K) is 0.2K. While this may not be statistically significant due to the length of the

simulations, this is not a negligible difference compared to the 1.5-2.0K Paris recommendations.

» Noted now in the text.

Section 5: The paragraphs in this section tend to be short and unconnected which makes it difficult to pull out the important messages of this study.

» This section has been modified to address this and other concerns. The goal was to try to state each result succinctly, but that perhaps did not work.

Figure 2: A different set of contour levels is needed to show the ozone changes.

» In response to this comment and that of the other reviewers, we have modified the contour intervals on the panels to be relative ozone changes. This does provide a much better picture.

Figures 3, 5, 6: The ozone panels don't add information here.

» As noted, these are now relative changes and they now provide more information.

Reply to Dr. Ponater

Âă Some comments on A. Gettelman et al. "Coupled chemistry climate effects from 2050 projected aviation emissions" Âă

(This is not meant as a full scale review but rather a collocation of ideas that occurred to me when reading through the paper. Nevertheless, I am confident that the comments express well-founded criticism.) Âă

» They do. Thank you very much for the comments.

The present paper fails to discuss evidence from previous work to an extent that makes it difficult to understand what is actually added here to current knowledge on the subject. Besides other papers I am particularly referring to Huszar et al. (2013), a basic study of future aviation impacts that the authors appear to have overlooked. Âă

» Thank you for highlighting this oversight. We have now noted the Huszar et al 2013 reference where appropriate. They discuss chemistry impacts (NOx) and contrail impacts, but not Aerosols.

1) The results given in section 3.1 all in all look scientifically and statistically plausible, yet they have apparently been presented before (Brasseur et al., 2015; Chen and Gettelman, 2016). However, I notice an inconsistency with re-spect to the contrail cirrus RF estimate for the 2006 baseline scenario between Table 2 (17 mW/m2) and Figure 1A (12 mW/m2). This is rather relevant, as the puzzling evidence that contrail cirrus RF increase more strongly over the years than fuel consumption would vanish, if the Table 2 value were taken as the starting value.Ăă

» Corrected. The 17 value is from a slightly different estimate in Gettelman and Chen 2013 that includes effects of water vapor beyond cirrus clouds. 13 mWm-2 is the correct value.

Recently, Forster et al. (2016) came up with a study indicating that radiative flux differences derived from free-running fixed SST simulations (I guess that's what "RESTOM" indicates in Figure 1) should amount to at least 100 mW/m2 in order to reach sufficient statistical significance levels. In case of nudged simulations (resembling the specified dynamics simulations in the present papers) the threshold value may reduce to 10% (Forster et al., 2016, p. 13), which appears to be consistent with the error bars in Figure 1. However, the error bars are clearly overlapping between the different scenarios at all time slices, indicating that the scenarios are statistically indistinguishable.Ăă

» Noted in the text.

2) I find the ozone pattern difference presentations from Fig. 2 a,b, Fig. 3 a,b, Fig. 5, Fig. 6, rather pointless. While they suggest large areas of statistical significance (for CESM almost over the whole troposphere), this remains un-convincing as contour lines are largely missing (those that are shown are mainly referring non-significant structures). Figure 8 of Huszar et al. (2013) offers a more satisfactory description,

clearly indicating that patchy stratospheric response patterns are insignificant, despite showing higher concentration difference values compared to the troposphere. It may, hence, be worthwhile to display relative differences, as in many earlier papers (e.g., Grewe et al., 1999) dealing with free running chemistry climate model simulation results.Âă

» These figures do indicate that the lower stratosphere shown is not significant. We have now changed the plots to relative ozone changes as suggested. This does present a much better picture and we think makes the figures more coherent.

3) The severe problems to assess the (statistical and physical) significance of temperature response patterns simulated from aviation effects have been re-ported before (e.g., Rap et al., 2010, Fig. 1a; Huszar et al., 2013, Fig. 10, Fig. 12). A point-by-point hypothesis test suggesting statistical significance in strongly confined regions may well turn out to be unfounded, if spatial correlation is accounted for. (Chaotic negative and positive side-by-side differences, as obvious in Fig. 2 c,d, Fig. 3 c,d, are always raising suspicions in this respect. I notice coherent regions of significance only in Figs. 3d, 4d, and 6f shown here.) Significant temperature response is more easily established for global means (Huszar et al., 2013, Fig. 9), but these are bypassed in the present paper. Sometimes, more sophisticated (multivariate) statistical tests have proved helpful to establish pattern significance (e.g., Sausen et al., 1998).

» We fully agree with your points. As suggested by reviewer #2, we have used the False Discovery Rate approach to better treat field significance (Wilks 2006), and this does eliminate the spuriously significant results (i.e. in the S. Hemisphere). We discussed significance of the global means, but have now made that more explicit with significance tests: the global means for CESM are (barely) significant. We are clear to not overstate the significance (barely significant at the 95% level).

4) In section 4.1.2 much text is devoted to allegedly large effects of aviation black carbon emissions without showing any results. To me this is absolutely unconvincing

as to underpin what is suggested by Figure 1b.Âă

» We have clarified the text here to flow better. We do specifically discuss further analysis here, and trace the effects back to the physical causes. Note that the effects are not in Figure 1B as they are only in the modified version of the model.

5) Given the general lack of statistical significant simulation results, I find large parts of the concluding section to be insufficiently covered by the results. In my opinion, the simulation strategy followed in this paper is only of very limited value for establishing reliable evidence on the relative importance of individual components in forcing a net aviation climate impact. Even in Huszar et al. (2013) statistical noise has made interpretation of their results problematic and I fail to notice any progress on this in the present paper.Âă

» We have looked again at the significance globally and using the false discovery rate approach from Wilks (2006,2016) as suggested. This provides a much better and robust approach for the results. The regional temperature changes are only significant in a few regions of aviation flight corridors, and the resulting global values in CESM are barely significant. We think this better treatment does validate the method of performing long coupled-climate simulations. We have endeavored to rewrite the conclusion section with these concerns in mind and the revised results to address this.

Adding on my main comments, I find the present paper to be written in a rather confusing manner. For example, the description of the simulations is scattered over three different sections (2.1, 2.2, 2.4) and it is not sufficiently recalled in the results section, which of the simulations are actually discussed. A special subsection (3.4.3) is dedicated to alternative fuel effects, yet those are partly addressed already in subsection 3.4.1.Âă

» We have reorganized the paper along the suggested lines. We moved the model sections together, and moved the scenario section, so that at least they are together. We also reference alternative fuel effects in 3.3.1 (only one sentence). We have added

discussion to the results as well that we think make this flow better.

References:Âă Forster, P.M., et al., 2016: Recommendations for diagnosing effective radiative forc-ing from climate models for CMIP6, J. Geophys. Res. 121, 12460-12475.Âă Grewe, V., et al., 1999: Impact of future subsonic aircraft NOx emissions on the at-mospheric composition, Geophys. Res. Lett. 26, 47-50.Âă Huszar, et al., 2013: Modeling the present and future impact of aviation on climate: an AOGCM approach with online coupled chemistry, Atmos. Chem. Phys. 13, 10027-10048.Âă Rap, A., et al., 2010: Estimating the climate impact of linear contrails using the UK Met Office climate model, Geophys. Res. Lett. 37, L20703.Âă Sausen, R., et al., 1998: Climate impact of aircraft induced ozone change, Geophys. Res. Lett. 24, 1203-1206.

Reply to Review #2

The paper presents results from two climate-chemistry-ocean models on the climate impact of future aviation. In particular the paper discusses the climate impact of aviation emissions on climate temperature changes.

Certainly, the topics are interesting and important and deserve careful investigations. An ocean-atmosphere coupled model with a high quality model simulating the fate of the aircraft emissions is essential for computing aviation climate change. Unfortunately, the paper is insufficient in many respects. The material presented, though not irrelevant, does not add enough new insight and results to the existing literature.

» We think there are some important insights in this paper that add to the literature. We think the revised manuscript with better statistical tests and revised conclusions makes a better and clearer contribution to the literature now.

This topic deserves a far deeper investigation and a technically better paper.

» It does deserve a deeper investigation. We have noted this now better in the conclusions and summary. We think these comments and those of the other reviewers have made this a technically better paper and we hope this will answer the reviewers'

concerns.

The paper reports quantitative values of the radiative forcing (RF) values for various aviation emissions and effects. The majority of these results are taken from earlier publications and given here with little or no discussion on the ranges of validity and discussions on the differences to results from other studies.

» This is not designed to be a comprehensive review, but is necessary to provide some consistency and a stand alone manuscript that makes sense. We have now explicitly noted where these different previous values come from in the text. We have tried to note previous studies as well to be more comprehensive.

Additions concern specific scenarios.

» Yes, that is one of the important features, and why it is necessary to add the background material.

Then the study reports various "nonlinear effects" from BC, including strong solar radiation absorption by BC from engine exhaust in cirrus, sulfur brightening of low level clouds, regional disturbances, advection effects, and surface temperature response. The simulations were performed with two different models. Here too, a large part of the results was published earlier (in the various cited papers by Jacobson, Chen and Gettelman). In fact, the paper has strong overlap with Brasseur et al. (2016), Jacobson et al. (2013), and Chen and Gettelman (2016), even repeating some of the numerical values in the tables and one of the already published figures.

» We feel it is necessary to add several missing points to previous work in the literature, which is why results are restated. They are also necessary to give the reader a sense of the overall picture. However, this work then goes beyond this significantly to show coupled model simulations and to compare them. This adds significantly by putting two disparate models in different contexts, and highlighting where previous results are confirmed or not in another complex and advanced modeling framework. This significantly builds on previous work by both model teams. We focus on our previous work for continuity, and because this is a model analysis, not a review.

The two models disagree in many respects, and the discussion mentions possible reasons, but the discussion remains qualitative and does not present new convincing evidence explaining the differences clearly.

» There is rarely ever a clear answer, but note we have done specific simulations to replicate one model with another for some of the key uncertainties. We try to highlight this better in the discussion and conclusions. We hae added some more quantitative analysis of the mechanisms for the aerosol effects in CESM for example. We naturally were not able to address everything, but we think we have a logical case that goes through the processes responsible for the major differences in sulfate and black carbon emissions between the models. It is difficult to ascertain all the differences between complex model systems. We feel we have gone farther than most. We have tried to bring up in the discussion now what these uncertainties mean. Again: these uncertainties and disagreement between models is a start of understanding of important issues, not the result.

Not surprisingly, the authors did not find statistically reliable results in this respect. This was to be expected, as discussed in other studies, because the disturbances are small compared to the inherent climate noise.

» Actually with some of the better statistics, we do clarify that there are regionally significant responses, and the global values are significant (barely). The revised statistical treatment to use field significance and the false discovery rate approach suggested was very helpful in this regard. We also now reference earlier work with coupled models (there is not much, but Huszar et al 2013).

For this reason, other authors either use enhanced disturbances or follow the idea of climate response models, which are quasi linear in the disturbances, with model parameters (inertia or heat capacities and climate sensitivities) fitted to full climate model

studies with enhanced disturbances. The possible inaccuracies of such approaches because of inherent nonlinearities are unavoidable.

» We did NOT want to do this because of these inaccuracies. Too many other studies have tried fitting or assumed linearity with large perturbations and scaling to address these issues, and our intent was to show what can be done with full response models. We have noted this in the conclusions better now. We also are able to better sort out the climate signals with revised statistical tests. We decided to try to run longer simulations (50 years) to better constrain the results.

It would have been interesting to see how linear or nonlinear the model responses are (see Rind et al., 2000).

» To some extent, this can be ascertained from figure 1, which does run simulations over different years.

As the authors mention themselves, none of the climate change results on regional temperature and ozone changes and surface temperature changes is strictly statistically significant.

» We have revised the tests, and we now have higher confidence that the regional responses are statistically significant as the new methodology removes a significant amount of climate noise. This is noted throughout the new results, discussion and conclusions. We now think some of the changes are statistically significant.

Even the global mean surface temperature changes remain in the statistical noise. (This seems to revise some earlier conclusions form the same data; e.g., Jacobson et a. 2013).

» That paper focused on the Arctic, and we do revise the results to note that those regional responses (particularly in the Arctic) are not seen in another model. However, we do find significant regional and global responses using the revised statistical tests. We have tried to make sure we clarify the results against earlier work. But this is an

important revision, and some confirmation to earlier work. The global mean surface temperature changes are just out of the statistical noise now.

The patterns of the simulation results do not look convincing. There is little systematic pattern in the mean responses. Many of the results just look random. Fig. 2b and 4 are examples.

» The revised statistical tests to use field significance to eliminate false discovery help eliminate this problem in the revised draft. We think the results are no longer random. For example, there is now no significance in the S. Hemisphere, and no significance anywhere for 5 years of coupled simulation.

I am not sure, whether the significance tests are reliable because based on local tests; see Wilks, D. S. (2016), "The stippling shows statistically significant grid points": How research results are routinely overstated and overinterpreted, and what to do about it, Bull. Amer. Meteorol. Soc., 97, 2263-2273, doi: 10.1175/BAMS-D-15-00267.1. This should be discussed.

» Thanks for the reference. We have not applied this test throughout, and it is very helpful for understanding the results, and we think it removes a lot of the spurious signals, and gives a much better sense of the significance of the results. As noted, we think this improves the paper quite a bit, and we thank the reviewer for this reference. We intend to use this method in the future. It seems quite valuable for reducing climate noise.

The suitability of the models for this study is not sufficiently justified. Since a global model with very coarse horizontal resolution (4 Å~ 5 degrees) cannot resolve plume dispersion of NOx and neither line-shaped contrail formation nor their merging into contrail cirrus, the results rely highly on the validity of the subgrid scale (SGS) models used. The present method gives no details on these SGS models but refers to previous publications.

»The plume dispersion (contrails) in GATOR-GCMOM is treated explicitly with a SGS model to enable effects to be handled even at low grid scale resolution. You are correct that NOx is not treated this way (this is noted). This is clarified in the text. A few further sentences have been added to describe the SGS and how it works.

When looking to some of the previous publications one finds a lot of adhoc simplifications. I agree that simplifications are unavoidable, but I see a lack of principle justification (e.g., on plume cross-sections), reflection of recent insight and data, and lack of validation of the SGS models with the observations that are now available from various studies.

» We have added further discussion of uncertainties and evaluation of GATOR-GCMOM against detailed models (i.e. a plume model) and CESM (against observations). The assumptions in CESM are analyzed parametrically in previous work, which we reference. Details of the sub-grid scale model in GATOR-GCMOM are also contained in previous work, and as noted above we now refer to them more explicitly.

The paper does not discuss the degree of agreement or disagreement with related contrail studies, e.g., in respect to contrail ice water content, optical depth, life times, cross-sections etc. Hence it is unclear of how good the SGS models are and how much the results change when the SGS model is changed.

» We have added references to evaluation for CESM and GATOR-GCMOM against observations, such as are available.

The assumption that the radiative properties of contrail cirrus are the same as those of normal cirrus clouds is highly questionable and has been overcome in other studies. We know since Minnis et al. (1998) that contrails remain observable at ages larger 10 h. Many further measurement results have been presented on this since then. Many findings indicate that aged contrails differ significantly from other cirrus. In particular they often contain higher concentrations of small ice particles, with impact on optical properties, sedimentation and life times.

» We have evaluated CESM against the Minnis et al data on contrail optical thickness (Pat Minnis was a co-author on the Chen et al 2012 paper). We now cite the validation of method against Minnis work for cloud optical thickness.

A prerequisite for contrail cirrus simulations is the suitability of the simulations of ice supersaturation (and temperature). See Irvine, E. A., and K. P. Shine (2015), Ice supersaturation and the potential for contrail formation in a changing climate, Earth Syst. Dynam., 7, 555–568, doi: 10.5194/esd-6-555-2015. It would be important to show how good the present models resolve temperature and ice supersaturation along aircraft flight tracks, e.g., compared to qualified reanalyzes or to in-situ measurements.

» The evaluation of ice supersaturation in CESM is in Chen et al 2012 as well and we note it explicitly in the revised text.

With respect to NOX and O3, I miss a discussion of the dispersion of NOx etc. from aircraft engines to grid scale which is known to cause nonlinear O3 changes since early studies in the 1990's, depending on the dilution model assumed. See, e.g., Paoli, R., D. Cariolle, and R. Sausen (2011), Review of effective emissions modeling and computation, Geosci. Model Dev., 4, 643-667, doi: 10.5194/gmd-4-643-2011.

» This is now discussed further in the text with references.

The question whether aviation NOx emissions cause a positive or negative or zero RF is still under debate. See Pitari et al. (2016), Radiative forcing from aircraft emissions of NOx: model calculations with CH4 surface flux boundary condition, Meteorol. Z., 23, doi: 10.1127/metz/2016/0776. The CH4 surface boundary condition seems to matter. The present study uses prescribed CH4 at the surface which likely has consequences for the results.

» This is now noted in the text (introduction). The positive RF associated with the short-term NOX-induced O3 production is to some extent offset by the negative RFs associated with aviation NOX emissions leading to a relatively large uncertainty associated with the overall net NOX-induced RF (Holmes et al., 2011). It is noted that this large uncertainty is also in part due to the different spatial and temporal scales associated with the positive and negative forcing (IPCC, 1999; Wuebbles et al., 2010). It is generally reported that the net effect is positive (IPCC, 1999; Holmes et al., 2011). This indicates the importance of accounting for the full suite of aviation NOx-induced RFs when reporting the net aviation NOX-induced RFs which have been often reported considering just the main forcing components (Lee et al., 2009 and Holmes et al., 2011).

Work from other teams is hardly mentioned. Differences in the results between this study and other studies (e.g. for NOx induced O3 and CH4 changes or contrail RF) are neither mentioned nor discussed.

» We discuss contrail estimates with other models further (as noted below) in response to this and other comments. We have added more references. We have also added a paragraph to the section on ozone on page 2 that discusses more fully some further studies on the chemistry and the complexity of the NOx, O3 and CH4 results, and potential differences between models.

As mentioned in a comment by M. Ponater, the studies by Olivié et al. (2012) and Huszar et al. (2013) are related to this work, and should have been discussed.

» They are now mentioned. Huszar does not include aviation aerosols, Olivié has a simple calibrated description of contrails.

The results and conclusions are not always clearly presented. One example: The authors relate (in the abstract and the conclusions) the non-local surface temperature signal from local radiative forcing to advection. The assumption that advection is the reason for nonlocal behavior is reasonable and not fully new (Ponater et al, Ann Geophys., 1996; Shindell and Faluvegi: Climate response to regional radiative forcing during the twentieth century, Nature Geosci., 2, 294-300, doi: 10.1038/NGEO473, 2009; see also Rind et al., 2000).

» Now cited in the text.

However, this paper does not bring any new argument to support this conclusion in this paper, except that the (noisy) temperature patterns exhibits downstream shifts. Is that worth mentioning as a new finding in the abstract?

» Yes, the finding is worth mentioning, because too many studies in the literature assume that regional forcing results in regional response. Noted in the text.

I think, this needs more analysis.

» Noted in the discussion.

The authors claim that heat absorption by BC from aviation is large enough to cause strong warming in contrails. They refer to Liou et al. (2013) in this respect. There is no doubt that BC does change absorption when present in sufficient amount. The quantitative results depend on the assumed BC mass (and effective sizes) of the soot and the mass of ice particles (and their sizes) and the fraction of ice particles containing soot particles. It would be important to check the mass budget of the BC in ice particles and see if this mass budget is consistent with the aviation BC emissions, the lifetime of cirrus and aerosol sinks.

» The mass budget of BC in GATOR-GCMOM has been evaluated by Jacobson et al 2011 against observations, and this has been noted in the text.

For example one could analyze from the model results the total ice mass in the cirrus clouds regionally or globally, convert that to cross-sections, and could compare this with the total BC mass from aviation in the same clouds. I would not be surprised if the mass or area fraction of BC turns out to be by far smaller than that for cirrus ice.

» The BC mass is going to be much smaller than the ice mass in the clouds. The BC contributes to anomalous absorption as described in Jacobson et al 2012 and in the text. We make this clearer in the revised manuscript. We did not mean to imply that BC would be a large mass fraction.

In spite of parallel studies by Righi et al. and Gettelman et al., I am not convinced that aviation sulfur emissions change low level clouds in any significant manner.

» Agreed. However, that is what the results of simulations indicate. We now add a more recent study using very different methodology by Kapadia et al 2016 that indicates the same thing.

I miss a careful and critical discussion of the amount and concentrations of cloud condensation (CCN) particles of reasonable sizes which could be contributed by aviation compared to the many other sources for CCN. Again, one could compute from the model results related statistics. To my understanding, most of the CCN in stratus clouds come from non-sulfur sources.

» We examined the perturbations in more detail and state the percent perturbations to liquid clouds by aviation in 2050 v. no aviation, as well as the locations (more in the sub-tropics). This is now discussed more fully in the text to back up the discussion of the mechanism.

I do not know of any single measurement showing that aircraft could indeed change water clouds by sulfur emissions. So, to me, this effect appears to be purely hypothetical.

» We agree with this and note the extreme uncertainty here. It does seem hypothetical until we can verify it. But the mechanism we propose is self-consistent in the model.

---

## Author Comment (AC2) · 20 Jul 2017

**Coupled Chemistry-Climate Effects from 2050 Projected Aviation Emissions**

Andrew Gettelman[1], Chih-Chieh Chen[1], Mark Z. Jacobson[2], Mary A. Cameron[2], Donald J. Wuebbles[3], Arezoo Khodayari[3,4]

[1]National Center for Atmospheric Research, Boulder, CO, USA
[2]Stanford University, Palo Alto, CA, USA
[3]University of Illinois, Urbana, IL, USA
[4]now at California State University Los Angeles, CA, USA

*Correspondence to*: Andrew Gettelman (andrew@ucar.edu)

**Abstract.** Analyses of the climate effects of 2050 aviation emissions have been conducted with two coupled Chemistry Climate Models (CCMs) including experiments with coupled ocean models. Results indicate that radiative forcing from projected aircraft emissions causes statistically significant (at the 95% level) climate signals on regional and global surface temperature for 2050 conditions. Significant signals emerge only after 20-50 years of simulation. The baseline 2050 aviation emissions scenario projects emissions ~5 times those in 2006. Simulations suggest a corresponding growth in the climate impact of aviation by 2050. Previous uncoupled experiments indicate positive radiative forcing from contrails reaches +80mWm$^{-2}$. Enhanced upper tropospheric and lower stratospheric ozone ($O_3$) due to aviation nitrogen oxide (NOx) emissions causes a radiative forcing of +60mWm$^{-2}$. Changes in methane ($CH_4$) lifetime induced by aviation are estimated to cause -25mWm$^{-2}$ of radiative forcing in 2050. Simulations indicate that moderate changes in water vapor emissions from changes in combustion efficiency will not have significant forcing. Effects due to particles (black carbon and sulfur) included in these calculations suggest an important role for black carbon (BC) in increasing contrail cirrus ice crystal number, leading to net warming. Sulfur emissions brighten clouds and provide a net cooling, but this is dependent on uncertain background sulfur levels. Thus alternative aviation fuels with reduced sulfur and BC may alter the future climate impact of aviation, but the magnitude is dependent on specific processes represented and the background state. Many regions with high regional aviation forcing do not experience net surface temperature changes because of advective rather than radiative driving of temperatures. Short-lived non-uniform aviation forcing will thus affect climate differently than uniform forcing in the coupled climate system.

**1. Introduction**

Aviation has several important impacts on climate (Wuebbles et al., 2007; and Lee et al, 2009, Brasseur et al., 2008 & 2015). Commercial aircraft currently emit ~2% (Wuebbles et al., 2007) of global fossil-fuel related emissions of carbon dioxide ($CO_2$). Aircraft also emit water vapor ($H_2O$), nitrogen oxides ($NO_x$), various organic gases, sulfur dioxide ($SO_2$), black carbon (BC) soot, other organic matter, and particulate sulfate. These chemicals may alter upper level cloudiness, atmospheric composition, and climate in several different ways.

For low enough temperatures and high enough exhaust humidity, aircraft emissions create condensation trails, or contrails (Schmidt, 1941; Appleman, 1953). If the ambient air is supersaturated with respect to ice, linear contrails may persist for minutes to several hours (Minnis et al., 1998). The optical thickness of existing cirrus may be enhanced by the additional nuclei resulting from contrails that have aged due to spreading, shearing, and sublimating (Minnis et al., 1998). Aircraft emission effects on ozone ($O_3$) and methane ($CH_4$) lifetimes may also have radiative feedbacks on climate (Brasseur et al 2015).

Recent work summarized by Brasseur et al (2015) has shown significant effects from aviation aerosols due to (a) aerosol absorption, termed 'direct' radiative effects and (b) modification of contrail and natural cloud microphysical properties by aerosols, or 'indirect' radiative effects. These effects may be as large or larger than the radiative effects of contrails themselves (Righi et al., 2013, Gettelman and Chen, 2013, Zhou and Penner, 2014). Aviation soot aerosol may change their own properties as they age (Heymsfield et al 2010). Soot inclusions within and between contrail and background cloud particles also create a cloud absorption effect by enhancing contrail burnoff (Jacobson et al., 2013).

Many studies have explored the total non-$CO_2$ (short lived) effects of aviation radaitive forcing. Lee et al. (2009) performed a review and meta-analysis and estimated aviation radiative forcing (RF) of 59 to 110 mWm$^{-2}$ for Ozone, -52 to -28mWm$^{-2}$ for $CH_4$, 37 to 55mWm$^{-2}$ for linear contrails, -12 to -18 for $SO_4$ (direct) and 9 to 13mWm$^{-2}$ for BC (direct). The sign of the net NOx effect on ozone is still uncertain (Pitari et al., 2016). This is mainly because the positive RF associated with the short-term $NO_X$-induced $O_3$ production is offset by negative RF associated with aviation $NO_X$ emissions, resulting in a relatively large uncertainty associated with the overall net $NO_X$-induced RF (Holmes et al., 2011). Different spatial and temporal scales associated with the positive and negative forcing is another reason for such a large uncertainty (IPCC, 1999; Wuebbles et al., 2010). It is generally reported that the net effect is positive (IPCC, 1999; Holmes et al., 2011), although recently Pitari et al (2016) reported a small negative number (-6.7 Wm2-) for the net effect induced by aviation NOx. Some of these differences in the sign of the net aviation-induced NOx effect are in part due to differences in model chemistry schemes and the treatment of physical processes, including transport and other factors (IPCC 1999; Stevenson et al., 2004; Kohler et al., 2008; Holmes et al., 2011; Olsen et al., 2013) and some due to considering a different suite of NOx induced effects.

This manuscript evaluates and assesses the radiative forcing, ozone perturbation and surface temperature changes from projected levels of aviation emissions for 2050 in two advanced General Circulation Models (GCMs) with common scenarios for 2050 aviation emissions. The study focuses on simulations with coupled ocean, land and ice models, similar to Huszcar et al (2013) and Olivié et al (2012). The goal is to determine statistically significant signals for small perturbations in coupled simulations. This work is not designed to be a review of the many aspects of the effects of aviation on climate and chemistry (e.g. recent review by Lee et al 2010), but rather a summary of two comprehensive coupled climate models. Most previous studies have not been conducted with fully coupled "climate-response" models with small aviation forcing. This is due to the complexity of models and to the high potential climate noise that makes it difficult to find small signals from aviation (e.g. Forster et al 2016). Our results largely validate this problem, and show that aviation signals, even for 2050 climate are significant but hard to identify in comprehensive models.

[revised manuscript text omitted]

**2.5 Statistical Significance**

Statistical significance was assessed at individual points using a 2-sided Student's t-test (e.g. Chervain and Schneider, 1976), estimating
the standard deviation of annual means from the simulations with one degree of freedom for each year. Statistically significant differences
are defined as those above 95% confidence. We also performed a paired sample t-test with autocorrelation on the difference of the annual
means and got essentially the same significance results (with a more formal treatment of autocorrelation). The significance in the maps
and zonal means is estimated using the False Discovery Rate (FDR) method to reduce spurious climate variability (Wilks 2016). The short
nature of the GATOR-GCMOM simulations (5 years) means that the standard deviations may be large (and the degrees of freedom small).
CESM simulations can be evaluated in a similar 5-year window, or with a longer 20-year window from the first 20 or last 20 years (31-
50) of 50-year simulations to explicitly address the issue of short and transient simulations. Forster et al. (2016) indicate that radiative flux
differences derived from free-running fixed SST simulations should amount to at least 100 mW/m2 in order to reach sufficient statistical
significance levels. In case of nudged or specified dynamics simulations the threshold value may reduce to 10% (Forster et al., 2016, p.
13). As noted below, this is consistent with our results.

**3. Results**

First we present general results for fuel burn from the scenarios and illustrate the potential for contrail formation. We then examine the
model-derived radiative forcing from contrails and other aviation emissions, including the radiative effects of chemical perturbations and
the effects resulting from aircraft emissions of soot and sulfur. Next we examine the model-derived chemical effects on ozone and methane
from NOx emissions. Finally, we analyze the resulting temperature response from the aviation perturbations in 2050 based on coupled
simulations. Note that the effect of alternative fuel is Scenario 2B v. Baseline for GATOR-GCMOM and Scenario 2 v. Scenario 1 for
CESM.

**3.1 Radiative Forcing in CESM**
* * *
Radiative forcing for contrails and from aviation-related aerosols (both direct and indirect effects) were assessed in CESM. Top of atmosphere net downward solar plus thermal-infrared irradiance changes (a surrogate for radiative forcing but not the same quantity) were assessed in both CESM and GATOR-GCMOM. For clarity we focus on CESM radiative forcing.

Figure 1 (adapted from Chen and Gettelman 2016) shows the evolution of the radiative forcing over time in CESM for different scenarios using SD simulations. The effects of water vapor only emissions (leading to contrail formation) are shown in Figure 1A, the effects of $H_2O + BC + SO_4$ emissions in Figure 1B and the effects of $BC+SO_4$ emissions only in Figure 1C, as the residual of panel B minus panel A. Note that since $H_2O$ emissions for Scenario 1 and Scenario 2 are the same, their RF in Figure 1A is the same. Figure 1A shows clear increases in contrail cirrus radiative forcing to 2050. While global emissions grow by a factor of 5, contrail cirrus RF may grow by a factor of 7 (13 to 90 mWm$^{-2}$). Note however that a warmer planet, using RCP4.5 meteorology (solid lines in Figure 1), lowers the resulting aviation RF relative to what it would be using 2006 meteorology (dashed lines in Figure 1), due to a reduction in contrail formation conditions at warmer temperatures (Chen and Gettelman, 2016). This assumes aircraft fly at the same pressure altitude in 2050 as today.

Global RF grows faster than the emissions because the regional pattern of emissions is changing. Emissions are rising faster in regions
like Asia, and in these regions the RF may increase more per unit emissions. The larger change in contrails may occur because there are fewer contrails in 2006 (there is a saturation effect) and/or the chance of forming contrails at flight altitudes is higher (since aircraft are always in the troposphere at low latitudes).

Figure 1C shows the impact of aerosols in CESM. In the scenarios with significant sulfur emissions (Baseline, red and Scenario 1, green)
there is significant decrease in TOA radiative forcing from aerosols (Figure 1C) that is anticipated to cause a near-surface cooling. Each line represents an average of 4 simulations (spread shown as error bars with symbols at the ends in Figure 1A and 1B).

Eliminating sulfur in Scenario 2 (blue) and Scenario 3 (purple) eliminates the negative radiative forcing from the indirect effects of aerosols (Figure 1C). The net forcing is then positive (Figure 1B) and similar to figure 1A. Scenario 3 has a small amount of additional water vapor
from aircraft. There is a small increase in contrail RF in 2050, on the order of 0-5% for a 5% increase in water vapor in Scenario 3, also seen in Figure 1B. The effect is small since most of the water in the contrails comes from the ambient atmosphere, not the fuel. The difference between scenario 2 and scenario 3 is not statistically significant.

Additional radiative forcing components are listed in Table 2, reprinted from Brasseur et al (2015). In particular, we have added CAM5 values of radiative forcing for 2050 for short-term effects of ozone, methane and water vapor for Scenario 2 (alternative fuels). We have also added available RF values for contrails and aerosols (Figure 1) from CESM-CAM5. Other values are as described in Brasseur et al (2015). New values are in bold, and show small effects of alternative fuels on minor species radiative forcing. The 2006 value for $O_3$ long and short term (~+30 $mWm^{-2}$) is larger than recent estimates by Pitari et al 2015 with 3 other models. The $CH_4$ value of -12 $mWm^{-2}$ is very similar to that of Pitari et al (2015). The reduction in sulfur and black carbon for alternative fuels in Scenario 2 changes short-term ozone RF by +12.0 $mWm^{-2}$ (17%), long-term ozone RF by -0.8 $mWm^{-2}$, methane by -2.2 $mWm^{-2}$, and water vapor by -0.5 $mWm^{-2}$. The short term ozone RF effect of the alternative fuel scenario (no BC and 50% S) is probably an 'indirect effect' on ozone that arises from changes in UTLS temperature induced by BC reductions, but the result may not be significant given the small changes in ozone in Figures

3A and B.

**3.2 2050 Aviation NOx-induced effects on Ozone**

The impact of aviation NOx leads to a direct short term ozone production that produces a positive RF (Brasseur et al 2015). A subsequent increase in OH then induces long-term reduction in methane ($CH_4$) that results in a negative RF, and a long-term small decrease in ozone generating negative RF (Wild et al., 2001). These effects are separated in Table 2, but combined in the figures. Some of the complexity and differences in multi-model results are discussed further by Brasseur et al (2015).

**3.2.1. Effects on ozone**

Figure 2A and B illustrate the simulated relative changes in zonally-averaged $O_3$ (in percent) due to aviation emissions for (A) GATOR-GCMOM and (B) CESM. Note that CESM uses years 30-50 of a long coupled simulation while GATOR-GCMOM uses the first 5 years of a 5-year simulation. Results for CESM at different averaging periods are in Figure 5, and are more similar between models. Statistical significance of zonal mean changes is indicated by black dots, and uses a 2 sided t-test with the FDR approach as described above.

Ozone radiative forcing is especially sensitive to the predicted ozone distribution in the Upper Troposphere and Lower Stratosphere (UTLS) (Lacis et al., 1990). GATOR-GCMOM (Figure 2A) projects a peak increase in ozone at cruise altitude between 45°N and 90°N of ~60 ppb, or over 30%. Which is partially significant due to the short simulation. Effects at higher altitudes likely depend on perturbations to the temperature gradient altering transport. However, due to variability of the stratospheric circulation they are not significant. Effects in CESM based on the first 5 years of a coupled simulation (Figure 5a) are similar in sign but only up to 20% (~20 to 30ppbv) and are also significant. They occur at a lower altitude however in CESM.

The effects of non-linear plume chemistry may affect the ozone results, as discussed by Paoli et al., [2011], depending on the dilution model assumed. Cameron et al, [2013] explicitly analyze this effect using a version of GATOR-GCMOM with a sub-grid scale plume model, and found 30% decreased $O_3$ production and a 10% change in NOx with a plume model.

**3.2.2 Effects on ozone with CAM5 using fixed meteorology fields**

CAM5 was also run with fixed meteorology fields for 2050 (based on meteorology from free running simulations). The NOx-induced changes in tropospheric ozone are complicated by a short-term increase in $O_3$ concentrations associated with a positive forcing, and a long-term reduction of $O_3$ concentrations tied to the aviation induced methane decrease. The long term-reduction is associated with negative forcing (Stevenson et al., 2004). The short-term $O_3$ forcing is one of the major contributors to the overall aviation forcing and dominates the net $O_3$ forcing (Lee et al., 2009). Since $CH_4$ mixing ratios at the boundary layer are fixed, the calculated changes in $O_3$ concentration are the short-term changes. This may have consequences for the results (Pitari et al 2016).

Radiative forcing effects from these aviation perturbations are illustrated in Table 2 for each of the scenarios in Table 1. Enhanced upper tropospheric and lower stratospheric ozone ($O_3$), termed "short-term $O_3$ forcing", due to nitrogen oxide (NOx) emissions contributes

~60mWm$^{-2}$ of radiative forcing. Changes to the methane lifetime induced by aviation are estimated to induce a radiative forcing of -25mWm$^{-2}$ in 2050.

**3.2.3 Alternative fuel effects on ozone**

Figure 3A and B show the effect of using an alternative fuel on ozone. Both CESM and GATOR-GCMOM indicate very small effects on ozone by switching from traditional fuel to alternative fuel with reduced BC and no sulfur. Values are not significant, except for some near surface values in the N. Hemisphere in CESM. The aviation NOx-induced effect on ozone in the presence of sulfur and BC appears to be the same as for standard fuel.

**3.3 Temperature Effects**

GATOR-GCMOM and CESM were run coupled to an active ocean to estimate the temperature and surface temperature response to
aviation emissions in 2050. The major motivation for running fully coupled climate simulations with GATOR-GCMOM and CESM is to
assess the effects of aviation radiative forcing due to chemistry, contrails and aerosols on surface climate.

**3.3.1 Surface Temperature changes**

Surface temperature changes are shown in Figure 4 for GATOR-GCMOM for the first 5 years for the effects of baseline emissions (Figure
4A) and the alternative fuel Scenario 2B (Figure 4B): none of these differences pass the significance test. Similar results are seen for
CESM simulations in the first 5 years (not shown). CESM is still adjusting to the forcing. CESM results over the years 31-50 for baseline
(Figure 4C) and the effects of an alternative fuel (Figure 4D) show more significant patterns in or adjacent to flight corridors. Jacobson et
al (2012) found with similar 20-year GATOR-GCMOM experiments 16% of the planet had 'significant' temperature changes (>95%
confidence level), including warming in the Arctic after a 20 year coupled experiment with present day aviation emission. In Figure 4 (and
in Jacobson et al 2010), regions of surface temperature change are not precisely co-located with the largest forcing, and are not significant
in 5 years of simulation using the FDR approach.  For baseline emissions, net cooling from aviation sulfate in CESM causes significant
Northern Hemisphere mid-latitude cooling (Figure 4C), consistent with the effect of the net negative radiative forcing for the baseline
scenario in Figure 1b. Cooling occurs in the deep water formation region around Greenland, and in the sub-tropical N. Atlantic, south of
the N. Atlantic flight corridor. The removal of sulfur in the Alternative Fuel Scenario 2 causes a reversal of this pattern (significant
warming), associated with an increase in net radiative forcing (Figure 1B: scenario 1 to scenario 2) due to the dominance of contrails. See
section 3.4.3.  Warming is significant in the N. Atlantic and N. Pacific, extending over Europe. But there is no significant warming over
the maximum forcing regions of Central Europe or Eastern N. America. The global temperature perturbations for the last 20 of 50 years
are $\pm 0.1$ $Wm^{-2}$ (negative with aerosol effects, positive with only contrail effects), barely significant at the 95% level (not significant at the
99% level).

We have also examined CESM initialized with a similar ocean structure to GATOR-GCMOM. The ocean initial condition is a uniform
temperature scaling with depth. The CESM simulation with such ocean initial conditions is not in balance, and over the course of the 5

years analyzed is still adjusting. This simulation was conducted to better compare with GATOR-GCMOM, and uses the same baseline emission scenario as GATOR-GCMOM. It highlights the transient nature of signals in the first 5 years of simulation. A case could be made that it is sufficient to look at the perturbation to the transient response between the two simulations, but the transient and longer term response seem different in CESM simulations because of the out of balance initial conditions from GATOR-GCMOM. Even after 50 years
with a small perturbation any the dis-equilibrium is smaller than the internal variability of the climate system (i.e., not significantly different than zero change).

**3.3.2 Vertical Temperature structure**

Figure 2 illustrates the impact of baseline aviation emissions on zonal mean temperature in GATOR-GCMOM (Figure 2C) and CESM (Figure 2D). Different time periods are shown: first 5 years for GATOR-GCMOM and years 31-50 for CESM. CESM is similar to GATOR-GCMOM over the first 5 years (illustrated in Figure 5). Over the first 5 years there is warming in the tropics and warming at flight levels, but cooling at higher altitudes and high latitudes. However, very little of this signal is significant given the variability in a 5 year simulation. For CESM some of the tropical warming and warming in the N. Hemisphere at flight altitudes is significant over the first
20 years of simulation (Figure 5) but this becomes a cooling over the last 20 years of a 50 year simulation due to the radiative cooling from increased aviation sulfur (Gettelman and Chen 2013). Initially CESM also has a N. Hemisphere UTLS warming due to short lived species and contrails (Figure 5), which as the SST cools over time due to aviation aerosols it becomes a cooling at the surface and in the UTLS (Figure 2D). This is not significant in the global mean.

In GATOR-GCMOM, BC is a strong contributor to the UTLS warming, and the reduction of BC in the Alt fuel scenario (Figure 3C) reduces warming relative to the baseline. The same is seen in CESM in the first 5 years of simulation (Figure 6). The UTLS warming due to aviation BC (and cooling from its removal with alternative fuels) occurs due to two factors. First, there is enhanced BC absorption upon coating by sulfate. Second, by enhanced BC absorption between contrail ice particles (Jacobson et al., 2013). Enhanced SW absorption results from hygroscopic growth of BC-coated aerosol particles, and BC absorbs not only solar radiation going down but also reflected
upward radiation, increasing the effect over snow and ice covered surfaces.

Most of the UTLS warming in GATOR-GCMOM (Figure 2C) appears to be due to BC. This was verified by conducting a sensitivity test with reduced BC only (not shown), which resulted in less tropospheric warming than the scenario shown. BC as pure aerosol warms the air by direct absorption of solar radiation. When BC is emitted from aircraft however it is coated by organic material, sulfate, and water. Coating increases BC absorption due to the optical focusing effect (Jacobson et al., 2013).

**3.3.3 Alternative Fuel Effects**

The reduction of BC in the alternative fuel scenario reduces the flight level warming in GATOR-GCMOM (Figure 3C) but increases it in CESM (Figure 3D and Figure 6), though neither is significant over a 5-year period. Over the last 20 years of a 50-year simulation (Figure 3D), the reduction of Sulfur in CESM alternative fuel Scenario 2 results in a significant warming near the surface in the N. Hemisphere as the cooling seen in Figure 2D is reversed when sulfur is removed in the alternative fuel scenario, as noted above. This is seen globally in Figure 1, and in the vertical temperature structure of Figure 3. The evolution of the temperature structure in the Alt Fuel scenario for CESM (Figure 6) highlights that the vertical structure of temperature changes quickly due to local radiative forcing, but then evolves in response to surface temperature effects. The BC effect is a local warming at altitude, while the brightening of clouds due to aviation sulfur affects the surface, which takes years to evolve. How this happens is best understood by looking in detail at the pattern of surface temperature changes.

**4. Discussion**

The major mechanisms by which aviation affects temperature differ between the two models. For GATOR-GCMOM, absorption due to aviation BC is important. For CESM, over land, contrail cirrus dominate the radiative forcing, while over oceans, this is negated by cooling due to sulfate brightening of low level liquid clouds. However, the initial responses of the models over the first 5 years of simulation are broadly similar. Below we attempt to understand and assess these differences.

**4.1 Sulfur**

Aviation sulfur affects climate in two ways. (1) Sulfate nucleates liquid cloud drops by deliquescence and activation of haze particles. (2) Sulfuric acid in the hot exhaust will condense onto BC particles to enhance the number concentration of contrail particles. Nucleation, (1), is unique to the co-emission of sulfur and BC (i.e., from aviation). Both models treat nucleation. Both models treat sulfur-BC mixtures, but GATOR-GCMOM treats sulfur-BC mixtures explicitly whereas CESM assumes BC and sulfate are internally mixed in the exhaust

Note GATOR-GCMOM BC distributions were evaluated by Jacobson et al (2011) against observations. In addition, sulfate coating of BC changes the absorption and ice nucleating properties of the BC aerosols in GATOR-GCMOM with both aviation and background emission. Aircraft aerosol sulfate is a small fraction of total aircraft-emitted sulfur, as much of the sulfur in both models manifests as $SO_2$, but that small amount of sulfate is a relatively large fraction of total emitted aerosol mass.

CESM produces a large cooling due to sulfate that is reversed when it is removed in Scenario 2 (Figure 1 and Figure 3 v. Figure 4). However, GATOR-GCMOM cools when Sulfate and BC are removed (Figure 3C v. Figure 4C). As noted in Section 3.3, the GATOR-GCMOM results are due to BC changes (BC warms). GATOR-GCMOM does not simulate large aviation sulfate indirect impacts on clouds. Possibly the main reason for the difference in sulfate cooling due to aviation in CESM (Figure 1) versus GATOR-GCMOM

(discussion in section 3.3.1) is that CESM predicts a much lower background (non-aviation) aerosol number concentration for particles >100nm, only 6% that of GATOR-GCMOM, so changes in emissions due to aircraft, have a greater impact on cloud indirect effects in CESM. CESM has larger (10%) perturbations to the number of Cloud Condensation Nuclei (CCN) in 2050 in N. Hemisphere mid-latitudes due to aviation. The increase in activated aerosols for liquid cloud formation in the N. Atlantic flight corridor is +30%, significantly increasing cloud drop number, and cloud liquid water. The maximum increase in drop number is in the sub-tropics where cooling is seen in Figure 4C. Results are consistent with advection and subsiding motion from higher aircraft cruise altitudes down to liquid cloud levels. These perturbations contribute to low-cloud brightening and cooling. Total Aerosol Optical Depth (AOD) levels (not shown) are much closer between CESM and GATOR-GCMOM because of similar numbers of small particles <100nm. In GATOR-GCMOM the higher background particle numbers and lower percent perturbation mean that aviation sulfur has less impact on clouds. However, previous work from Righi et al (2013) treats aerosols similarly to CESM, and finds similar cooling effects to CESM. Righi et al. (2013) did not treat cloud absorption effects of BC aerosols emitted by aircraft, and did not examine climate response, only radiative forcing. Kapadia et al (2016) using a very different model and methodology also found aviation aerosols could impact liquid clouds below.

**4.2 BC (Soot)**

GATOR-GCMOM has a large impact from the effects of enhanced BC absorption at altitude, due to (a) direct effects of sulfur coated BC, (b) cloud absorption effects of BC in contrail ice, (c) semi-direct effects, and (d) indirect effects on contrail ice number. GATOR-GCMOM treats these effects, including the coating of BC particles by sulfate, ammonium, water, and other aerosol constituents, enhancing cloud absorption effects of BC aerosols. At the high super-saturations seen in aircraft exhaust, all BC particles should activate to small liquid

The impact of sulfur aerosols indicates that significant changes to aviation sulfur emissions will have large effect on the climate response to aviation emissions. .

drops that eventually freeze to form ice crystals in the plume. These may then coalesce into larger drops. The BC inclusions result in a significant absorption increase (Jacobson et al 2011; Jasobson et al 2012).

The CESM BC treatment is similar, with some important differences. The Modal Aerosol Model (MAM) in CESM does mix BC and
sulfate aerosol and simulate enhanced absorption due to internally mixed aerosol BC, but MAM in CESM does not treat enhanced absorption by contrail particles containing BC. CESM does not treat enhanced absorption by aerosol BC between non-contrail cloud particles at the relative humidity of the cloud or within cloud particles. CESM does not assume an initial particle number based on BC, but rather an 'aged' or coagulated particle number based on coagulated particles (10 microns in size). The BC aerosol absorption enhancement due to mixing of BC aerosol with sulfate is included in CESM, but there are significant differences in the treatment of
particle sizes and optical properties between the grid scale model of CESM and the plume model in GATOR-GCMOM. However, we do see similar initial warming in CESM as in GATOR-GCMOM in the baseline scenario (Figure 2 and Figure 5) before effects of sulfur cooling in CESM alter surface temperature.

To analyze the differences in BC treatment, CESM experiments were conducted to bring CESM assumptions closer to those of GATOR-
GCMOM. We tested any differences in the size distribution of BC between CESM and GATOR-GCMOM, altering CESM simulations to reflect smaller BC size and higher numbers. CESM makes no distinction between background BC and aviation BC, and CESM assumes 0.1% efficiency of ice nucleation for all BC particles (Gettelman et al., 2010). The plume processing that occurs in GATOR-GCMOM first allows for coagulation that reduces BC particle number significantly. Then, nearly 100% of the resulting BC particles can activate to form contrail particles in the highly-supersaturated contrail plume that exists immediately behind an aircraft (Jacobson et al., 2011). To
bridge this difference CESM simulations were performed where the contrail ice particle number concentration was set to the aviation soot number, both in a specified dynamics (to get radiative forcing) and coupled simulation. This results in much higher ice crystal numbers in contrails in CESM, similar to what GATOR-GCMOM would produce in a contrail. In this test, the positive cloud forcing in SD simulations from contrail cirrus in CESM increases by a factor of 4 and there is a net warming that is more similar to GATOR-GCMOM in magnitude. The warming is large enough in this simulation to reduce contrail formation due to the warming (burnoff) as in GATOR-GCMOM.

Thus it appears that the BC soot aerosol particles emitted by aircraft increase the number of contrail ice crystals, increasing the optical thickness, for a net warming (as the enhanced longwave absorption dominates over the enhanced shortwave reflectance). Previous studies showed similar effects. For example, Wong et al. (2013) found using laboratory experiments and modeling studies that soot was internally

From this analysis, it appears that the increased contrail number due to BC aerosols and absorption creates a larger warming in GATOR-GCMOM. Similar effects can be produced in CESM with similar assumptions of higher contrail numbers from activation on BC and limited coagulation.

**Moved down [1]:** Previous studies showed similar effects. For example, Wong et al. (2013) found using laboratory experiments and modeling studies that soot was internally mixed in contrail ice particles and Liou et al. (2013) found a resulting increase in cloud heating rate due to these internal mixtures.

**Moved (insertion) [1]**

mixed in contrail ice particles and Liou et al. (2013) found a resulting increase in cloud heating rate due to these internal mixtures. This seems in contrast to significant cooling from soot effects on cirrus clouds noted by Zhou and Penner (2014)

**4.3 Ozone**

Both models show ozone concentration enhancement in the upper troposphere and lower stratosphere (9-12km) in the N. Hemisphere (e.g., Figure 2A from GATOR-GCMOM) from the effects of aviation emissions. CESM effects are seen at lower altitudes (in the troposphere, not in the lower stratosphere). CESM shows similar effects for simulations run as a CTM, without differences in meteorology or temperature. Neither model shows significant effects from the alternative fuels scenario on the changes in ozone concentrations (Figure 3A and 3B). However, these effects are seen in the CESM uncoupled SD experiments with a lower noise threshold (Table 2). Ozone changes are consistent with NOx and NOy changes in CESM. Increases in ozone concentrations are consistent with previous work assessed in Brasseur et al (2015).

**4.4 Surface Temperature**

Both GATOR-GCMOM and CESM performed 2050 simulations of surface temperature changes. GATOR-GCMOM has a global mean warming in the simulations relative to aerosol emissions, but it is not significant in a five-year simulation. In CESM, signals are not significant after 5 years, only after 50 years. Most of the effect is in the atmosphere and the surface takes time to react. In general, surface temperature effects of aviation emissions in 2050 are statistically significant only in regions of maximum emissions and forcing over oceanic flight corridors, and barely significant globally. Similar results were found for present day emissions by Jacobson et al. (2012) with 20-year GATOR-GCMOM experiments. The cooling from indirect effects of aviation sulfur in CESM results in a global average mean temperature change of -0.11K after 50 years (Figure 4C), significantly different from zero at the 95% leevel. This means that the short term aviation signal from contrails and aerosols, even at 5-8 times current levels in 2050, have a barely significant globally-averaged surface temperature signal. The alternative fuel scenario (scenario 2, Figure 4D) is globally warmer (+0.2K) than the baseline scenario (Figure 4C) in 2050 due to the large positive radiative forcing, and this is globally and regionally significant.

Regional patterns of temperature change in Figure 4 are correlated with short term aviation radiative forcing. For CESM simulations with strong cooling from sulfate, there is cooling over the N. Atlantic, particularly in regions where deep water is forming in the ocean (Figure

Deleted: The effects of soot can significantly change the climate impact of aviation. To further constrain the BC soot effects for climate purposes, it would be necessary to characterize better the near term evolution of ice particles and try to constrain better the particle size distributions in contrails. Further analysis of existing observations of such size distributions against the model size distributions could be conducted, but that for the moment is beyond the scope of this work. .

4C). There is a reduction of this cooling (warming) if effect of sulfur is removed with an alternative fuel (Figure 4D). Note that the warming and cooling in the coupled system occur not necessarily where the largest forcing is (over N. America and Europe), but where the climate system is most sensitive to perturbations (the sea ice edge and the region of deep water formation). Similar findings for 20[th] century emissions were noted by Shindel and Faluvegi (2012). Also, the climate system takes time to react to the forcing at the surface, particularly over the ocean.

**5. Conclusions**

This study has analyzed the climate effects of 2050 aviation emissions using the same 2050 emissions scenarios with two very different climate-chemistry modeling systems. The study highlights new work and particularly work with models coupled to an active ocean to elucidate surface temperature responses. Small perturbations to the coupled climate system result in regionally significant responses that are barely significant globally, but detectable after 50 years of simulation. Furthermore, we have attempted comparisons between two coupled models (GATOR-GCMOM and CESM). Specific simulations have been performed to replicate one model with another for some of the key uncertainties and processes, which we highlight below. Key conclusions concern aviation radiative forcing, the importance of indirect effects of sulfur and BC emissions, and resulting temperature responses assessed with coupled simulations in this work. The two models provide a consistent picture, when taken into context.

**5.1 Radiative Forcing**

Simulations indicate significant growth in the climate impact of aviation that occurs with projected increases in fuel use. Fuel use increases by a factor of 3 to 5 relative to 2006, while radiative forcing from contrails increases 13 to 90 mWm$^{-2}$ (7 times 2006 forcing). There are also large Ozone radiative forcing responses in 2050. Short term $O_3$ forcing is estimated at ~60mWm$^{-2}$, and changes to the methane forcing induced by aviation-caused methane lifetime changes are estimated at -25mWm$^{-2}$ in 2050 (based on differences with and without aviation). Ozone responses in 2050 to aviation emissions result from increases in ozone in flight corridors of 20-60ppb (10-30%). These effects from aviation may be large (particularly in otherwise pristine regions), depending on the future background emissions.

**5.2 Sulfur and BC Indirect Effects**

Simulations in both model systems indicate that effects of aviation emissions, through interactions with aerosol particles, can be larger than the effects of aviation $CO_2$ or aviation $H_2O$ emissions that cause contrails (Figure 1). The main particle emissions affecting radiative forcing are sulfate and BC soot.

Sulfur rapidly evolves in the atmosphere into sulfate aerosols that can interact with liquid clouds. The effect of this process will be net cooling, but mostly over oceans. Because CESM predicts lower background aerosol concentrations of larger aerosol particles, changes in emissions due to aircraft increase aerosol loadings by a larger fraction in CESM resulting in a greater impact on indirect effects in CESM. As such, CESM predicts more cloud-enhanced cooling than GATOR-GCMOM of as much as -150 mWm$^{-2}$ in CESM in 2050 (Figure 1C). The physical mechanism in CESM is aerosol formation at flight altitudes, which advects in subsidence regions into the sub-tropics where it can affect liquid clouds at low altitudes. This effect seems speculative, but has been seen significantly in other studies (Righi et al 2013, Kapadia et al 2016) using different models and methods.

GATOR-GCMOM indicates that BC, coated by sulfate and efficiently activating to form contrail particles increases contrail absorption (heating), while CESM does not simulate sulfate coating BC in contrail plumes to increase activation rates. However, there is still warming at altitude in CESM either from BC or from contrails. When large BC numbers are assumed to efficiently create contrail particles in CESM, there is better agreement in results between the two models. Thus, BC emissions appear in both models, if treated similarly, to cause increases in contrail absorption (net heating).

As a result of these sulfur and BC effects, both model simulations indicate that the use of alternative fuels that substantially reduce sulfur and BC emissions from aviation may significantly alter the future climate impact of aviation. The pathways indicated by the two modeling systems are similar. These effects are large in both models examined, but depend on the nature of the processes represented (BC enhanced absorption in GATOR-GCMOM) and the background state (lower sulfate in CESM). The magnitude of potential positive forcing from BC effects still needs to be determined and better constrained. The BC and sulfur effects are uncertain, and further comparisons between models and observations of contrail aerosols would be valuable and important to better constrain these effects. Specifically, to further constrain the BC soot effects for climate purposes, it would be necessary to characterize better the near term evolution of ice particles and try to constrain better the particle size distributions in contrails. Further analysis of existing observations of such size distributions against the model size distributions could be conducted, but that for the moment is beyond the scope of this work.

Indirect Sulfur and BC effects would be mitigated with alternative fuels. Reduced fuel burn (Scenario 1) reduces impacts across all different aviation components. Alternative fuels with lower BC and no sulfur (Scenario 2 and 2B) significantly reduce indirect effects of aerosol sulfur and BC while not affecting contrail formation very much. The result would shift the balance of aviation radiative forcing effects but with reductions of positive forcing for BC reductions, and reduced negative forcing for sulfur reductions. There are still large uncertainties in this finding that need to be better assessed by more fully analyzing how the models treat aviation BC soot and sulfate aerosols.

**5.3 Surface Temperature**

The global surface temperature effects of aviation in 2050 are small, and barely significantly different than zero in CESM. Effects in the Arctic seen in GATOR-GCMOM (Jacobson et al. 2013) are also seen in some CESM simulations over short periods of time, but not in longer CESM coupled simulations. This means that some of the transient differences may be a product of the slightly different evolution of feedbacks in the transient GATOR-GCMOM and CESM simulations. Effects in CESM are seen over longer periods (Figure 4C and 4D) in response to large 2050 aviation forcings. Arctic effects are seen in CESM only in simulations set up with an ocean state out of balance. However, the effects of temperature were not examined with CESM with background aerosol levels similar to those in GATOR-GCMOM. Thus high latitude effects, particularly surface temperature responses, should be studied more, but will require long equilibrated simulations with coupled models.

Regional radiative forcing for 2050 is concentrated over regions of high emissions, either directly over high traffic land regions of Asia,

N. America and W. Europe, or over oceanic flight corridors of the N. Pacific and N. Atlantic. But, regional radiative forcing does NOT translate into regional surface temperature responses even when regional forcing reaches $2Wm^{-2}$. The heat budget of middle and high latitudes is not controlled locally, but has a large transport component. Over regions of the largest positive aviation radiative forcing over continents, temperature may be more due to advection than local radiative fluxes. Persistent short term regional forcing impacts the climate system where the system is most sensitive (i.e., regions of the sea ice edge and deep water formation) rather than where forcing is largest.

The impact on the surface takes time to develop after radiative forcing is integrated for many years. Thus, short-lived and non-uniform radiative forcing (e.g., from aviation aerosols and contrail effects) may have a climate response per unit of forcing very different from long-lived and uniform forcing (such as from aviation $CO_2$). Local effects on temperature may not be correlated with local radiative forcing. Long coupled simulations are needed to be able to observe these effects, even for 2050 emissions whose forcing is several times

Deleted: Ozone responses in 2050 to aviation emissions indicate increases in ozone in flight corridors of 20-60ppb. These effects from aviation may be large (particularly in otherwise pristine regions), depending on the future background emissions. Short term O$_3$ forcing is estimated at ~60mWm$^{-2}$, and changes to the methane forcing induced by aviation-caused methane lifetime changes are estimated at -25mWm$^{-2}$ in 2050 (based on differences with and without aviation).

that of 2006. Further work with other coupled (climate-response) models would be valuable to evaluate or falsify the pathways and the magnitude of the effects identified here in two models.

**Acknowledgements**

This work was made possible by the support of U.S. Federal Aviation Administration Aviation Climate Change Research Initiative (ACCRI) grants to Stanford, NCAR and University of Illinois. Thanks to Andy Heymsfield and Rangasayi Halthore for comments, and to Rangasayi Halthore for continued support and encouragement. NCAR is supported by the U.S. National Science Foundation.

**TABLES**

**Table 1:** Description of Emissions Scenarios used in this work.

| Scenario List | |
|---|---|
| Baseline | Baseline Fuel Burn (technology improvements not included) |
| Scenario 1 | Reduced Fuel Burn (includes assumed technology improvements) |
| Scenario 2 | 'Alternative Fuel'; Reduced Fuel Burn, no Sulfur, 50% BC |
| Scenario 2B | GATOR-GCMOM Baseline 'Alt Fuel' Scenario |
| Scenario 3 | Scenario 2 with 5% increased $H_2O$ |

**Table 2** Comparison of aviation emissions component specific RF (mW/m$^2$) for 2006 and 2050 baseline and alternative scenarios (From Brasseur et al., 2015, Table 4) with new results from this study added in bold.

| Scenario | Fuel burn (Tg) | NO$_x$ (Tg N) | O$_3$-S UIUC CAM5* | CH$_4$ UIUC CAM5* | Long Term O$_3$ UIUC CAM5 | Water Vapor UIUC CAM5 | Contrails NCAR CAM5 | Aerosols NCAR CAM5 |
|---|---|---|---|---|---|---|---|---|
| 2006 -Base | 188.1 | 0.812 | 36.5 | -12.3 | -4.5 | -2.6 | 13 | -38 |
| 2050-Base | 902.8 | 3.950 | 143 | -59.7 | -20.3 | -12.5 | **83** | **-160** |
| 2050-S1 | 514.4 | 1.570 | 70.5 | -28.3 | -9.4 | -5.9 | **72** | **-107** |
| 2050-S2 | 514.4 | 1.570 | **58.5** | **-25.6** | **-8.8** | **-5.4** | **72** | **0** |

**Figures**

[Figure]

**Figure 1: Radiative forcing in CESM from (A) H$_2$O emissions alone (contrail cirrus), (B) H$_2$O +BC + SO$_4$ emissions and (C) The residual effect of BC and SO$_4$ estimated from (B) – (A). Solid lines are for the RCP 4.5 scenario with different meteorology each year; dashed lines in panel (A) are using 2006 meteorology. Since scenario 1 and scenario 2 have the same H$_2$O emissions, they**
**have the same RF in (A). The baseline aviation scenario is in red, scenario 1 is green, scenario 2 is blue and scenario 3 is purple.**

[Figure]

**Figure 2: Zonal mean change in (A,B) ozone and (C,D) temperature due to baseline 2050 aviation emissions (v. no aviation emissions). Panels A,C for GATOR-GCMOM and B,D CESM. Black dots show regions of statistically significant perturbations using the False Discovery Rate method as described in the text. Based on first 5 years of simulations for GATOR-GCMOM, and years 31-50 for CESM.**

[Figure]

Figure 3: Zonal mean change in (A,B) ozone and (C,D) temperature due to Alternative Fuel 2050 aviation emissions (v. 2050 aviation emissions). Panels A,C for GATOR-GCMOM and B,D CESM. Black dots show regions of statistically significant

**perturbations using the FDR method as discussed in the text. Based on first 5 years of simulations for GATOR-GCMOM, and years 31-50 for CESM.**

[Figure]

**Figure 4: Surface temperature changes in (A, B) GATOR_GCMOM and (C, D) 50 year CESM coupled simulations. A and C are for baseline emissions (v. no emissions) and B and D are for an alternative fuel scenario. Black dots show regions of statistically significant perturbations using the FDR approach as described in the text.**

[Figure]

**Figure 5: CESM simulations for different time periods. Baseline – No Aviation. (A,B,C) dO3, (D,E,F) dTs. (A,D) First 5 years, (B,E) First 20 years, (C,F) Years 31-50. Black dots show regions of statistically significant perturbations using the FDR method as described in the text.**

[Figure]

**Figure 6: CESM simulations for different time periods. AltFuel – Regular fuel (Scenario 2 – Scenario 1) (A,B,C) dO3, (D,E,F) dTs. (A,D) First 5 years, (B,E) First 20 years, (C,F) Years 31-50. Black dots show regions of statistically significant perturbations using the FDR approach as discussed in the text.**

**2.3 AEDT Emissions/Scenario**

Simulations are conducted using a common set of emissions scenarios. Emissions used in this study come from the Aviation Environmental Design Tool (AEDT) emission inventory (Barrett et al., 2010; Olsen et al., 2012; Brasseur et al., 2015). The AEDT dataset is an hourly inventory of global aircraft emission mass of ten emission species over a 1° x 1° latitude-longitude mesh with a vertical spacing of 150 m in the year of 2006. 2006 data are chorded (individual flight tracks), and 2050 data are scaled and gridded.  For GATOR-GCMOM, individual flight track 2050 aircraft emissions are obtained by extrapolating 2006 emissions in the same flight paths to 2050 by scaling the current year 2006 flight data by the ratio of 2050 to 2006 emissions. Emission indices from Barrett et al., (2010) are used for different species to determine particulate emissions. Both models use greenhouse gas concentrations in 2050 from the Representative Concentration Pathway 4.5 scenario (RCP4.5).

Different scenarios are listed in Table 1. The Baseline scenario assumes no operational or technology improvements. Scenario 1 has reduced fuel burn because of assumed technology improvements. Scenario 2 assumes the same fuel burn of Scenario 1 with an 'Alternative Fuel' (Alt Fuel) that has no sulfur and 50% less BC (soot) emissions. Scenario 3 assumes Scenario 2 (reduced fuel burn with no sulfur and 50% BC) with +5% higher aviation $H_2O$ emissions from the engine exhaust. These scenarios were implemented in CESM. GATOR-GCMOM used 2050 baseline emissions, and a modified scenario 2 that assumes fuel burn of the Baseline scenario and an Alt Fuel version that has no sulfur and 50% less BC. This scenario (run only by GATOR-GCMOM) is called 'Scenario 2B'.

The baseline emissions scenario has a fuel burn in 2050 that is 5 times that of 2006 emissions. Over E. Asia, the 2050 emissions are nearly 8 times those of 2006. 2050 emissions changes are lower over the U.S. (2 times 2006) and Europe (4 times 2006). The reduced fuel burn scenario (Scenarios 1 and 2) is only 3 times 2006 emissions, with 4 times 2006 Asia, and 2 times 2006 for the U.S. and Europe.

, Simulations indicate significant growth in the climate impact of aviation that occurs with projected increases in fuel use. Fuel use increases by a factor of 3 to 5 relative to 2006, while radiative forcing from contrails increases 12 to 90 mWm$^{-2}$ (7 times 2006 forcing). The baseline scenario (the high end of this range) assumes no technology improvement and is likely an unrealistic upper bound. It is shown here for comparison purposes. Regional radiative forcing for 2050 is concentrated over regions of high emissions, either directly over high traffic land regions of Asia, N. America and W. Europe, or over oceanic flight corridors of the N. Pacific and N. Atlantic. Regional temperature change happens where the climate system is most sensitive near regions of large forcing, but not always coincident with largest forcing.The diurnal timing of contrail effects of aviation may mean that there are some regional effects not treated here, though diurnal cycle effects (Chen and Gettelman, 2013) are smaller than uncertainties in the model formulations or emissions.